# Deciphering cell lineage specification of human lung adenocarcinoma with single-cell RNA sequencing

Zhoufeng Wang[1,2,3,11], Zhe Li[4,11], Kun Zhou[5,6,11], Chengdi Wang[1,11], Lili Jiang[7], Li Zhang[1], Ying Yang[1], Wenxin Luo[1], Wenliang Qiao[8], Gang Wang[2], Yinyun Ni[1], Shuiping Dai[9], Tingting Guo[1], Guiyi Ji[10], Minjie Xu[4], Yiying Liu[4], Zhixi Su[4], Guowei Che [6✉] & Weimin Li [1,2,3✉]

Lung adenocarcinomas (LUAD) arise from precancerous lesions such as atypical adenomatous hyperplasia, which progress into adenocarcinoma in situ and minimally invasive adenocarcinoma, then finally into invasive adenocarcinoma. The cellular heterogeneity and molecular events underlying this stepwise progression remain unclear. In this study, we perform single-cell RNA sequencing of 268,471 cells collected from 25 patients in four histologic stages of LUAD and compare them to normal cell types. We detect a group of cells closely resembling alveolar type 2 cells (AT2) that emerged during atypical adenomatous hyperplasia and whose transcriptional profile began to diverge from that of AT2 cells as LUAD progressed, taking on feature characteristic of stem-like cells. We identify genes related to energy metabolism and ribosome synthesis that are upregulated in early stages of LUAD and may promote progression. MDK and TIMP1 could be potential biomarkers for understanding LUAD pathogenesis. Our work shed light on the underlying transcriptional signatures of distinct histologic stages of LUAD progression and our findings may facilitate early diagnosis.

[1] Department of Respiratory and Critical Care Medicine, Frontiers Science Center for Disease-related Molecular Network, West China Hospital, Sichuan University, Chengdu, Sichuan, China. [2] Precision Medicine Research Center, West China Hospital, Sichuan University, Chengdu, Sichuan, China. [3] Research Units of West China, Chinese Academy of Medical Sciences, West China Hospital, Chengdu, Sichuan, China. [4] Singlera Genomics Ltd, Shanghai, China. [5] Department of Thoracic Surgery, The First Affiliated Hospital, Zhejiang University School of Medicine, Hangzhou, Zhejiang, China. [6] Department of Thoracic Surgery, West China Hospital, Sichuan University, Chengdu, Sichuan, China. [7] Department of Pathology, West China Hospital of Sichuan University, Chengdu, China. [8] Lung Cancer Center, West China Hospital Sichuan University, Chengdu, Sichuan, China. [9] Center of Gerontology and Geriatrics, West China Hospital, Sichuan University, Chengdu, Sichuan, China. [10] Health Management Center, West China Hospital, Sichuan University, Chengdu, Sichuan, China. [11] These authors contributed equally: Zhoufeng Wang, Zhe Li, Kun Zhou, Chengdi Wang. ✉email: cheguowei_hx@aliyun.com; weimi003@scu.edu.cn

Lung cancer is the leading cause of cancer-related deaths worldwide, and the most prevalent subtype is lung adeno-carcinoma (LUAD)[1]. LUAD is thought to progress most often from atypical adenomatous hyperplasia (AAH) to adeno-carcinoma in situ (AIS), then to minimally invasive adenocarci-noma (MIA), and finally to overt invasive lung adenocarcinoma (IA)[2]. Several studies have tried to understand LUAD progression based on the frequencies of genetic alterations during progression to malignancy[3]. Nonetheless, recent studies only reported increase in the frequency of genetic alterations during cancer progression to malignancy[4,5], but the underlying molecular event and level of cellular heterogeneity remain unclear.

The lung is composed of multiple cell types. Basal, club, and ciliated cells are predominant in the proximal airway. The alveolar wall contains alveolar type 1 (AT1) and type 2 (AT2) cells[6]. Studies in mice have suggested that LUAD may originate in AT2 cells, bronchioalveolar stem cells (BASCs), or club cells[7]. Some of these mouse studies have utilized lineage tracing tech-nologies, but similar experiments could not be performed in human[7–9].

Single-cell RNA sequencing (scRNA-seq) can provide a com-prehensive, unbiased catalog of cellular diversity within lung tissue[10–13]. Studies based on scRNA-seq have provided infor-mation about the roles of stromal cells in a few LUAD subtypes[14], and about cellular reprogramming in metastatic LUAD[15,16]. However, the diversity of cell states at different points during progression, especially during the early stages, and what roles different cell types play in progression are largely unknown.

Here, we perform scRNA-seq on matched tumor and normal samples from 25 patients in four different histological stages of LUAD. Our results provide the single-cell transcriptome atlas for all major subtypes of LUAD. They further suggest that AT2 cells dedifferentiate into a cell stem-like state, which helps initiate and maintain tumor progression. We further identify genes related to energy metabolism and ribosome synthesis that may be helpful for diagnosing LUAD in early stages. Taken together, our findings contribute to the understanding of how LUAD progresses at the cellular and molecular levels.

## Results

**Characterization of cellular heterogeneity across four LUAD subtypes.** A total of 52 freshly resected lung specimens were collected from 25 patients in different histologic subtypes of LUAD (3 AAH, 5 AIS, 9 MIA, and 17 IA) (Fig. 1a and Supple-mentary Data File 1), along with 18 adjacent normal lung tissues from a distal region within the same lobe, which served as con-trols. Eight of the 25 patients presented multiple nodules (Fig. S1). We selected nodules that showed pure ground-glass character-istics by computed tomography (CT), to reduce nodule hetero-geneity in AAH, AIS, and MIA samples. Tumor specimens were cut along the largest diameter, and half were processed for intraoperative freezing and paraffin embedding, while the other half were carefully cut along the inner side of the nodule edge in order to minimize contamination of normal tissue. All samples were evaluated by two pathologists to determine pathologic diagnosis and tumor cellularity (Supplementary Data File 2 and Supplementary Data File 3).

For each specimen, we rapidly digested the freshly collected tissues to generate a single-cell suspension, and the isolated live cells were used directly (Fig. S2), without enrichment steps, in scRNA-seq on the 10x Chromium platform. We characterized the transcriptome of 140,556 cells from patients P1-P22 at single-cell resolution using the V2 kits and validated our results on a separate dataset of 127,923 single cells from multiple nodules in patients P23-P25 processed using the V3 kits (Supplementary

Data File 4). We also performed whole exome sequencing (WES) or whole genome sequencing (WGS) on a subset of 26 tumor samples that have available DNA, and identified mutations in nine canonical driver genes, including *BRAF*, *EGFR*, *ERBB2*, *HRAS*, *KRAS*, *MAP2K1*, *MET*, *NF1* and *ROS1*[3], while driver mutations in *EGFR* and *KRAS* mutation were detected in only six tumor samples (Fig. S3).

To identify distinct transcriptional profiles of different cell populations, we performed dimensionality reduction and unsu-pervised cell clustering using the Seurat package[17]. We identified cell clusters based on their key marker gene expression and assigned them to 16 major cell types (Fig. 1b–d, and Fig. S4a), comprising epithelial types (ciliated, club, basal, AT1, AT2, and AT2-like cells) and stromal types (endothelial cells, fibroblasts, lymphocytes and myeloid cells). Analysis of normal and tumor epithelial cell types revealed a group of cells closely resembling that of AT2 cells (AT2-like cells) and were enriched in malignant cell population (Fig. 1b–d). Comparisons between normal and malignant cells in the four histologic stages showed that most normal cells were immune cells, and that each cell cluster contained cells from multiple patients (Fig. 1c, Fig. S4b, and Supplementary Data File 5). The frequency of some cell types varied significantly during progression from normal lung to AAH, AIS, MIA and then IA (Supplementary Data File 5 and Fig. S4c–d). For example, the enrichment of T and B lymphocytes as well as the decline in natural killer (NK) cells and granulocytes during tumor progression, suggesting activation of adaptive immune responses, which is consistent with a previous report that immune evasion may have started as early as the preneoplastic stage[18]. These results illustrate a high level of transcriptomic heterogeneity within LUAD, could be at least partially modulated by the surrounding microenvironment during progression.

Given previous studies suggesting that AT2 cells are the origin of LUAD[9,17], we calculated correlation coefficient of gene expression levels between different cell types and found that AT2-like cells correlated strongly with AT2 cells (Supplementary Data File 6), which was confirmed by comparison of gene expression profiles (Fig. S5) and the correlation coefficient among epithelial cell types (Supplementary Data File 7). Notably, we observed very small percentage of cells expressing cell prolifera-tion markers, so we opt not to correct for the cell cycle effect (Fig. S6a, b). Overall, this analysis revealed AT2-like cells were associated with malignant cell population, and AT2 cells are likely the origin of LUAD.

Recent advances in scRNA-seq have allowed researchers to closely examine the diversity of molecular and transcriptional states of lung cancer cells in the IA stage (Supplementary Data File 8), but molecular events in early-stage LUAD remain poorly understood. Here we reanalyzed the data published from two LUAD patients of the total eight lung cancer patients in a previous study[14], based on the same normalization and filtering parameters in Seurat package (Fig. S7). We identified the same 16 cell clusters as in our dataset. In fact, both datasets were highly consistent in the cell types detected. However, our dataset captured additional rare cell populations such as ciliated cells, lymphocytes, and early-stage AT2-like cells that were absent from the previous study (Fig. S7a, b). Most of the epithelial and stromal cell types identified in our primary dataset of 22 patients were validated in a group of 127,923 single cells from additional 3 LUAD patients (Supplementary Data File 1 and Fig. S7c–e).

**Characterization of epithelial cell lineages across different histologic stages of LUAD.** Lung epithelial cells have been stu-died extensively due to their role in lung cancer and various

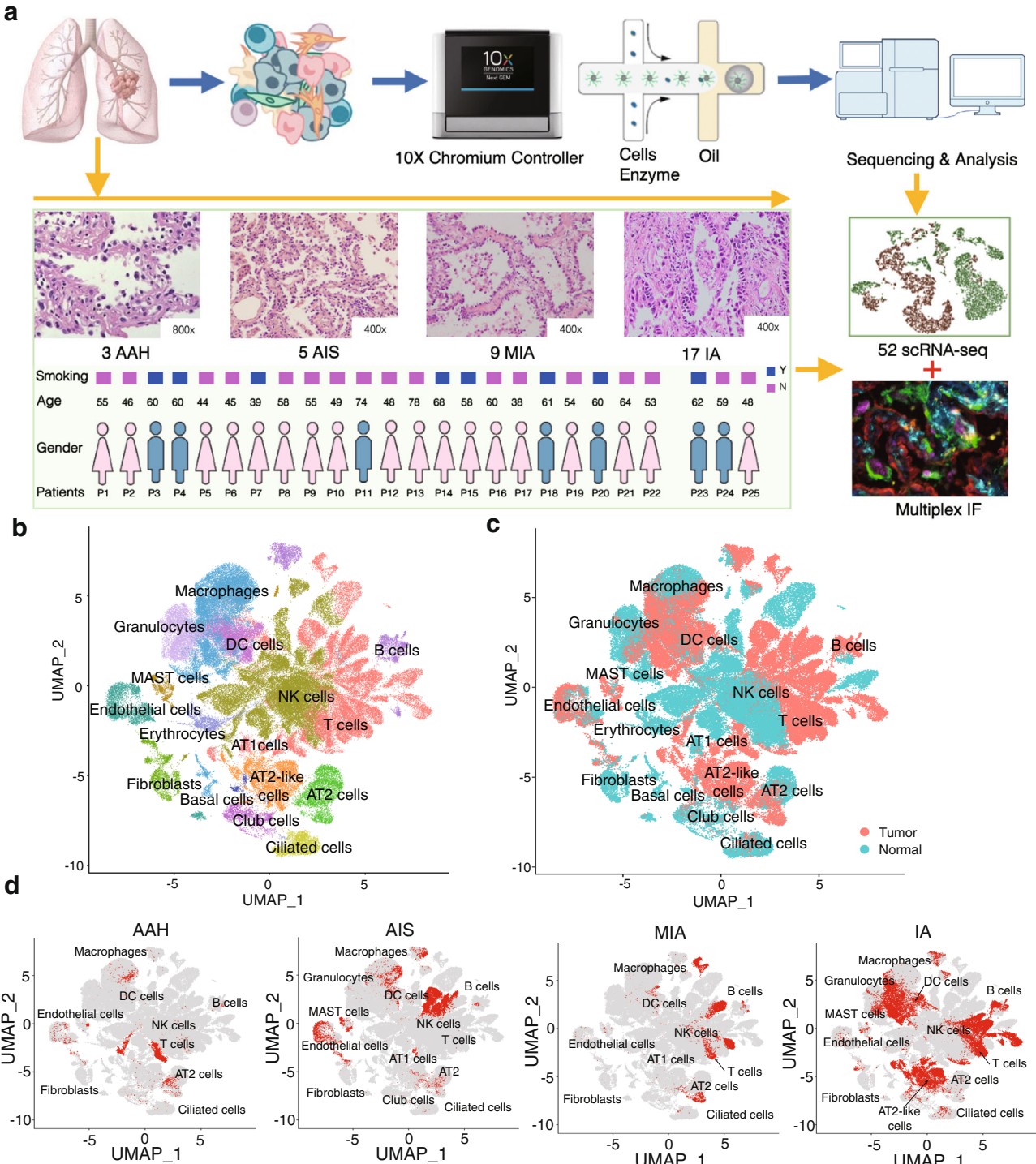

**Fig. 1 Overview of the single-cell transcriptomic profile of LUAD samples. a** Schematic experimental workflow for the study. A total of 34 freshly resected lung tumor specimens were collected from 25 patients, and normal lung tissues were collected from 18 patients as controls. Samples from patients 1 to 22 were processed using 10X Chromium V2 kits, while samples from patients 23 to 25 were used as the validation dataset and processed using 10X Chromium V3 kits. **b** UMAP visualization of 16 major cell types identified and color-coded by their associated clusters. **c** UMAP visualization of the 16 major cell types identified and color-coded by tumor or normal lung origin. **d** UMAP visualization of 16 major cell types identified and color-coded by histologic stages. LUAD: lung adenocarcinoma; UMAP: Uniform Manifold Approximation and Projection.

pulmonary diseases such as asthma and fibrosis[19]. Here we compared the transcriptomes of normal and tumor epithelial cells from four histologic stages of LUAD. We identified 15,984 epithelial cells and grouped them into 10 subclusters. (Fig. 2a–c and Fig. S8). Based on the expression profile of known markers, we found that the epithelial cell atlas mainly comprised of AT1 cells

(*PDPN* and *AGER*), AT2 cells (*HHIP*, *SFTPC* and *SFTPA*), club cells (*SCGB1A1* and *CP*), basal cells (*Krt5* and *TP63*), ciliated cells (*FOXJ*1 and *CCDC78*) and AT2-like cells (*MDK* and *SFTPB*) in Fig. S5 and Fig. 2c. As expected, normal epithelial cells showed five sub-populations expressing well-defined epithelial markers of the six forth mentioned cell types (Fig. 2b, c). Interestingly, the

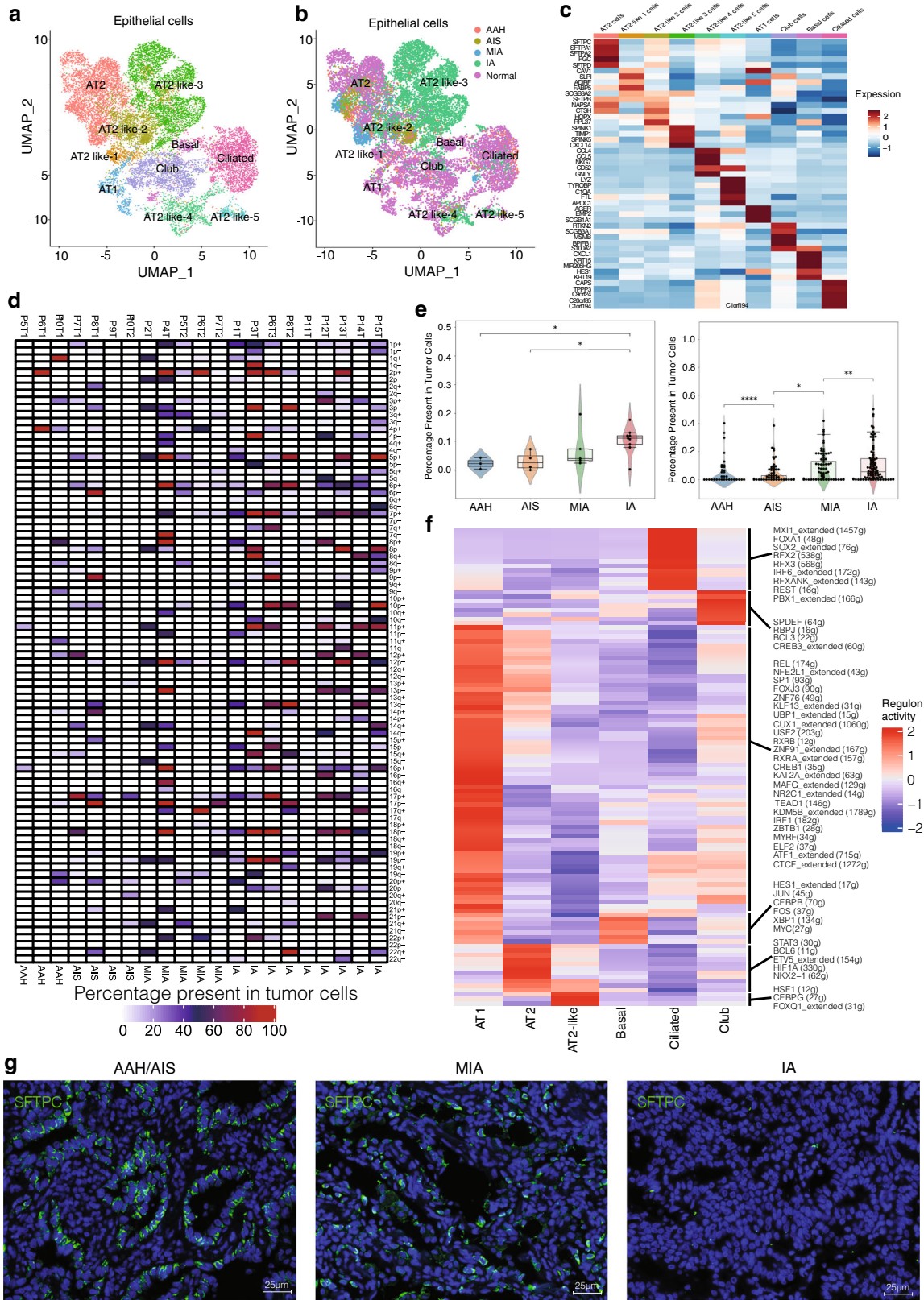

AT2 cell marker SFTPC was highly expressed in normal and early-stage tumor tissues (AT2 and AT2-like 2 cell clusters), but expression diminished in IA-stage tissues (Fig. 2c), consistent with the loss of AT2 cell identity during tumor progression

previously reported in genetically engineered mouse models (GEMM)[13].

In tumor tissues, epithelial cells may contain residual normal cells in the malignant tumor cell population. Here we used

**Fig. 2 Characterization of epithelial cell lineages across different stages. a** UMAP visualization of epithelial cell sub-clustering, color-coded by the identified cell subtypes. **b** UMAP visualization of epithelial cell sub-clustering by histologic stages. **c** Heatmap of selected marker genes in each cell cluster subtype. **d** Summary plot of the inferred CNV profiles from each of the 15 patients; CNVs were annotated by the chromosome arm in which the CNV events were calculated. Chromosomal amplification (red) and deletion (blue) were extrapolated in each chromosomal position (columns) across the single cell (rows). The color bar represents the assigned cell-type signature for each cell. **e** Violin plot of CNV percentage present in tumor cells at different stages. Each dot represents the average CNV (amplification or deletion) percentage in each stage (left) and the CNV events in chromosome arm in each stage (right). Mann–Whitney two-sided test is used to test the significance of CNV levels between different tumor stage categories. (AAH stage: $n = 3$, AIS: $n = 4$, MIA stage: $n = 5$ and IA stage: $n = 9$; Left: AIS vs IA $p$-value is 0.0253, AAH vs. IA $p$-value is 0.0420; Right: AAH vs AIS $p$-value is 2.898e-05, AIS vs. MIA $p$-value is 0.0348, MIA vs. IA $p$-value is 0.00799; *: $p < 0.05$, **: $p < 0.01$, ***: $p < 0.001$, ****: $p < 0.0001$). **f** Heatmap of gene expression regulation by transcription factors using SCENIC for the epithelial cells. **g** Protein fluorescence immunostaining for SFTPC in human tumor samples from the representative tumor stages (i.e., AAH to IA). Each staining in panels come from three samples. Nuclei were stained blue (DAPI). Scale bars: 25 μm. CNV: copy number variation; SCENIC: Single-Cell Regulatory Network Inference and Clustering. Source data are provided as a Source Data file.

inferCNV on epithelial cells to distinguish between tumor and normal cells based on large-scale somatic Copy Number Variation (CNV) events (Fig. S9 and Fig. 2d, e). We used the cell types from normal lung tissues as a healthy reference to estimate the CNVs of four histologic stages tumors. Chromosomal amplification (red) and deletion (blue) were mapped to each chromosomal position (columns) of the AT2-like cells in Fig. 3d. The results suggested that AT2-like cells obtained large-scale chromosomal CNVs in IA stage (Fig. S9). Specifically, the AT2-like cells showed a gradual increase of chromosomal gain/loss events throughout cancer progression, while other epithelial cell subtypes had little or no somatic CNVs (Fig. 2e and Fig. S9). Therefore, these results represent strong genetic evidence supporting AT2 cells as the origin of LUAD.

Next, we employed single-cell regulatory network inference and clustering (SCENIC)[20] to assess the differences in expression levels of transcription factors (TFs) in epithelial cells. We found that RFX family motifs were highly activated in ciliated cells, and that MYC and HES1 were highly upregulated in basal cells (Fig. 2f). The TFs SPDEF, RBPJ, and CREB3 were activated in club cell cluster. Canonical AT1 TF such as MYRF was expressed in AT1 cells[17]. Notably, we found that the lung fate TFs Nkx2-1 and the AT2 cell identity TFs Etv5 were enriched in the normal AT2 cell cluster. In contrast, the TFs HSF1, CEBPG and FOXQ1 were highly upregulated in AT2-like cells. HSF1 overexpression is common in lung cancer and correlates with tumor angiogenesis, while HSF1 activation in early-stage lung cancer cells and stroma are associated with poor outcome[21]. The TF CEBPG correlates with antioxidant and DNA repair genes in airway epithelium and is associated with predisposition to lung cancer[22]. Thus, our findings are consistent with the literature, and suggest that TFs of AT2 and AT2-like cells help drive LUAD tumorigenesis and progression.

To confirm the expression pattern of our epithelial markers, we performed immunofluorescent staining to determine the abundance and spatial localization of AT1 cells (AGER), ciliated cells (FOXJ1), club cells (SCGB1A1), basal cells (Krt5) and AT2 cells (SFTPC, Nkx2-1, SFTPA) (Fig. S10 and Table S1). Normal lung alveolar and bronchi tissues were double-stained with anti-AGER and anti-SFTPC antibodies. AT1 and AT2 cells were localized mainly in peripheral alveoli, while club cells, ciliated cells and basal cells were distributed mainly on the bronchial surface as previously described[14]. In LUAD tissues, expression of the SFTPC was observed in MIA tumors, but less than half of the AT2 cells expressed SFTPC compared to that of AAH (Fig. 2g and Fig. S11a). In IA stage tumors, SFTPC expression was almost undetectable, and the alveolar structure was not recognizable. Consistent with the reduced SFTPC expression, Nkx2-1 and SFTPA staining was lower in IA stage tumors than in early-stage tumors (Fig. S10b and Fig. S11a). Therefore, canonical AT2 cell marker gene expression significantly decreased during cancer

progression. Consistent with our findings, another recent study reported loss of Nkx2-1 and SFTPC expression during LUAD progression in mouse model[23]. Our results, together with the literature, suggest that AT2 cells may be the origin of LUAD and we therefore designate them as AT2-like cancer cells.

**Transcriptional trajectory of AT2 cells**. To identify the key molecular events governing the cell-fate transition during progression from normal to cancer cells, we selected cell clusters that closely resemble those of AT2 cells and AT2-like cancer cells, and then tracked the gene expression changes along the trajectory from AAH, AIS, MIA and finally to IA. We performed pseudotime analysis based on Monocle2 and observed non-random expression patterns (Fig. 3a–c). The transcriptional states in the trajectory revealed progression-associated changes in tumors. Tumor cells at early stages (AAH or AIS) gathered on one end, while cells from late-stage tumor tissues (MIA or IA) were on the other end (Fig. 3a).

We identified 283 differentially expressed genes that exhibited dynamic expression over pseudotime ($q$-value < 0.05) and classified them into four groups (groups 1-4). Then, we ordered these genes along pseudotime and reconstructed a diffusion map (Fig. 3b). The expression profile of group 1 showed relatively quiescent self-renewing AT2 genes (with high level of WIF1 inhibiting WNT), and high expression of several stem-like cell transcription and differentiation genes (LAMP3, MUC1)[10,24]. By contrast, the gene expression profile in group 2 resembled the start of dedifferentiation, involving upregulation of the RNA biogenesis processing (RPS) family[25,26] and the mitochondrial factors MT-ND4 and MT-ND2[27]. The expression profile of group 3 reflected inflammatory responses triggered by cytokines (FOSB, NFKB1) as well as expression of the EMT-related gene Vmentin[28]. Lastly, the genes in group 4 were involved in extracellular matrix organization (Tissue Inhibitor Matrix Metalloproteinase 1, TIMP1)[29] as well as cell–cell signaling and regulation of cell migration (S100A4, VEGF)[30]. We also identified several genes previously linked to cancer progression, such as midkine (MDK), SOX4 and LYZ[1,31]. Although each of these expression patterns emerged at a different time, they all persisted in tumors once occured, such that more advanced tumors contained a greater diversity of cells in different states.

We next examine the changes in marker gene expression along the pseudotime (Fig. 3c). JUN, TIMP1 and MDK were highly expressed at the IA stage, whereas LAMP3 were substantially diminished as LUAD progressed. Elevated levels of MDK, a product of lysine decarboxylation, were also identified as one of the most important features for discriminating IA stage LUAD[31]. TIMP1 have been reported to regulate metabolism in metastases by activating the PI3K/Akt pathway[29], which we confirmed by immunostaining of our tissue samples (Fig. S11b, c). Our results

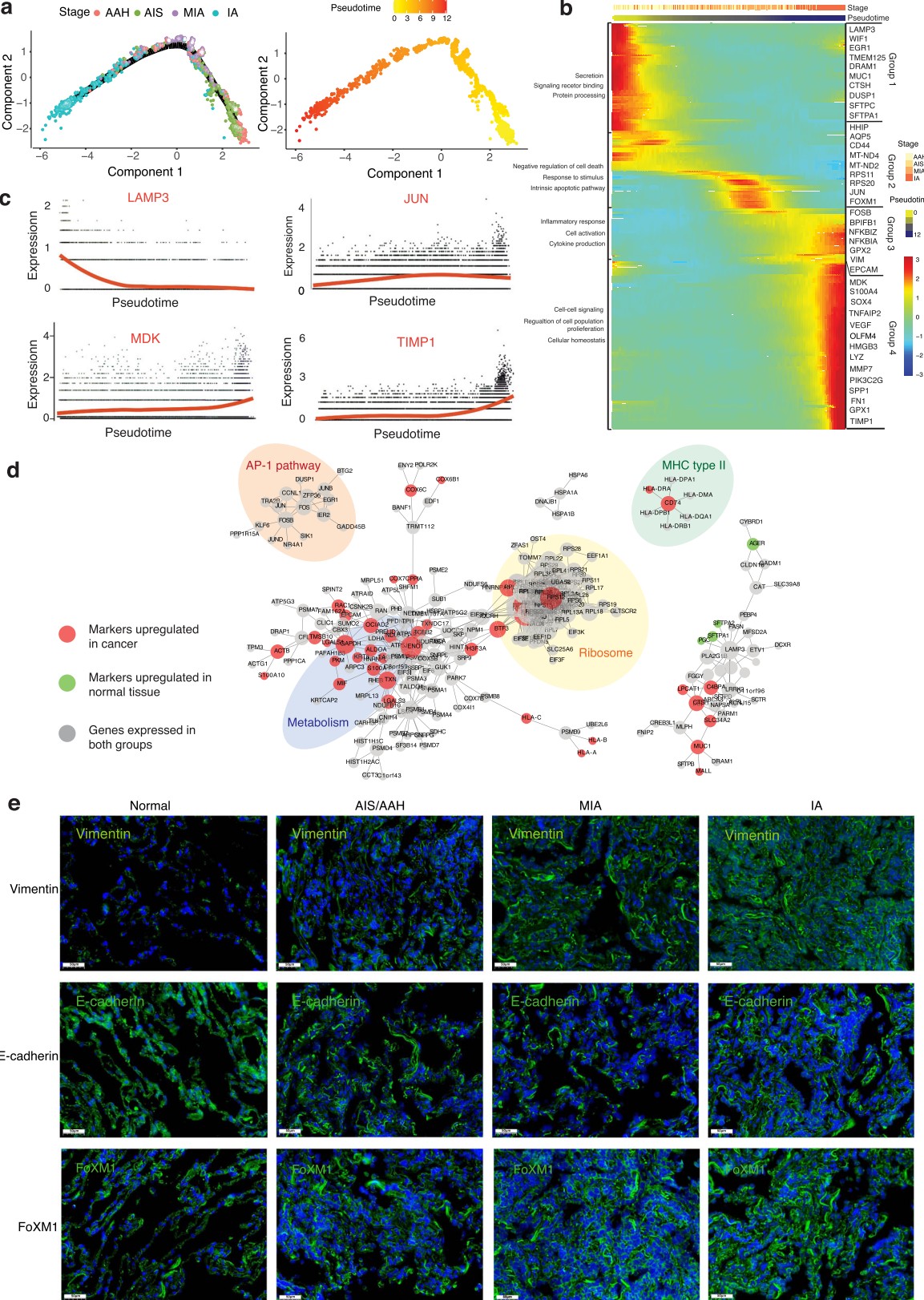

**Fig. 3 Transcriptional trajectory analysis of AT2 cells. a** Trajectory analysis of AT2 and AT2-like cell clusters. Cells were color-coded by histologic stages or pseudotime. AT2: alveolar type 2 cells. **b** Heatmap showing 283 differentially expressed genes arranged in pseudotemporal patterns. Gene Oncology terms from the enrichment analysis revealed biological functions of cells in the four groups indicated. **c** Representative gene expression levels of different marker genes. The size of each dot represents relative expression levels. **d** Gene-gene interaction networks between marker genes in AT2 and AT2-like cell clusters. **e** Fluorescent in situ hybridization staining for *Vimentin*, *E-cadherin*, and *FoXM1* in normal tissues and tumor tissues from different stages. Each staining in panels come from three samples. Scale bars: 50 μm, nuclei (DAPI) are stained blue.

highlight *MDK* and *TIMP1* as potential biomarkers for understanding LUAD pathogenesis.

After filtering out genes expressed in fewer than 10% of all cells, we examined genetic interactions between 3613 genes in the AT2 and AT2-like cell clusters (Fig. 3d). We found that genes upregulated in normal tissues are association with pulmonary surfactant-associated protein and homeostasis. In contrast, the genes upregulated in tumor are involved in metabolism, ribosomal activity, or MHC class II molecule expression, which suggests that these activities are essential during tumor progression. Notably, the genes significantly downregulated in tumor cells are related to immune activation, supporting a previous finding that LUAD cells suppress immune responses[14].

**Loss of AT2 features and gain of stemness are associated with LUAD progression**. Multiple mouse studies have suggested that AT2 cells retain a self-renewal activity of stem cells that helps to drive cancer progression[32,33]. Consistent with this notion, we uncovered an AT2-like cell cluster whose transcriptional profile closely resembled that of AT2 cells initially, but gradually lost AT2 cell transcriptional identity while retain features of the lung epithelial lineage. These AT2-like cells, which expressed many stemness genes (*CD44*, *IFI27* and *S100A4*), were present in tumors throughout LUAD progression (Fig. 3b). This observation suggests that LUAD progression involves a loss of AT2 features for the lung lineage and the emergence of an alternative dedifferentiated, stem-like state. These results are consistent with the idea that several diseases, especially cancer, involve dedifferentiation of committed epithelial cells[34].

Dedifferentiation of epithelial cells may play a role in the promotion of epithelial-mesenchymal transition (EMT)[28]. To investigate the prevalence of EMT in LUAD progression, we used RNA fluorescence in situ hybridization (RNA-FISH) to examine the expression levels of the mesenchymal marker *Vimentin*, the epithelial marker *E-cadherin* and *FOXM1*, a pro-stemness transcription factor associated with proliferation of kidney and ovarian tumors[35,36] (Fig. 3e and Fig. S12a). Expression of *E-cadherin* decreased in conjunction with increased *Vimentin*. We also found increased expression of *FOXM1* as LUAD progression. Our results suggest that *FOXM1* also could be a driver of dedifferentiation and proliferation in LUAD[37]. These findings were further supported by tissue immunofluorescence and bioinformatics analysis of known epithelial and fibroblast markers.

We next used indirect immunofluorescence staining to validate our scRNA-seq findings based on levels of Ki67, Vimentin and VEGF protein. Ki67 expressed in all phases of the cell cycle except $G_0$. Interestingly, the staining intensity in our study showed an increase in Ki67 expression (gray) from normal to AAH/AIS stages, however, there was no significant changes during progression (Fig. S12b and Fig. S13). This finding corroborates the clinical practice of using Ki67 expression to make treatment decisions in LUAD[38]. Immunostaining also showed that expression of the EMT marker Vimentin and the angiogenesis marker VEGF increased during LUAD progression.

Wnt contributes to stem cell self-renewal and lineage-specific differentiation in diverse tissues[39,40]. Wnt signaling was found to be amplified by engaging the leucine-rich repeat-containing G-protein-coupled receptor Lgr5, which is a marker for stem cells in multiple epithelial tissues and can drive lung adenoma progression in mouse model[39]. RNA-FISH revealed a higher *Lgr5* expression in all four histologic subtypes of LUAD than in normal tissues (Figs. S12c and S14). We also detected two WNT mediators of *GPX2* and *OLFM4* (Fig. 3b), which can be activated by *Lgr5* to drive tumor progression[41]. On the other hand, the

expression of the stem-like genes *IFI27* and *S100A4* increased as LUAD progressed (Fig. 3b). Our results suggest that the stem-like transcriptional signature correlates with increased tumorigenic potential. Therefore, we speculate that AT2 cells dedifferentiate into a stem-like state in which they initiate and maintain tumor progression.

**Characterization of stromal cells in coordinating tumor microenvironment during LUAD progression**. Analyzing the stromal cells associated with tumors could provide deeper insights into lung cancer biology[14]. To investigate stromal cell dynamics in the tumor microenvironment (TME), we examined the single-cell transcriptomes of endothelial cells (ECs), fibroblasts, lymphocytes, and myeloid cells from normal and tumor tissues in the four histologic stages of LUAD. We detected 3,925 ECs and five clusters based on marker genes (Fig. 4a–c and Figs. S15–S16). We next attempted to identify marker genes for each of these clusters and to assign them to known endothelial cell types. These clusters included tip-like cells, tumor ECs, stalk-like cells, endothelial progenitor cells (EPCs) and lymphatic ECs. Most of the EC clusters belonged to normal tissues and could be assigned to known vascular cell types[14]. For example, lymphatic ECs were enriched in normal tissues (Fig. S16a). Tumor ECs were observed in the tumor tissues of all four histologic stages. Tumor ECs in early-stage tumors strongly expressed *PLVAP*, *GSN*, and *TSC22D1*, which are relevant to the development and cell-fate commitment of ECs[15]. To gain more biological insights underlying these cell states, we used Gene Set Enrichment Analysis (GSEA) to compare expression profiles between tumor and normal ECs (Fig. 4c and Supplementary Data File 9). The top enriched signature in tumor ECs included Myc targets and the interferon (IFN) pathway. The c-Myc protein is essential for tumor angiogenesis, glycolysis and oxidative phosphorylation, all of which promote vessel sprouting[42]. Upregulation of IFN-γ and IFN-α pathways is associated with inflammatory responses. So, these processes may play a role in ECs biology. The endothelium represents the primary interface between circulating immune cells and the tumor, this may help explain how ECs contribute to LUAD[43].

Fibroblasts are known to be heterogeneous, but their heterogeneity in LUAD progression is unclear[44]. We detected six clusters of fibroblasts, including fibroblast-like cells, normal fibroblasts, smooth muscle cells, lipofibroblasts and myofibroblasts (Fig. 4d–f and Fig. S15). Expression profiles are consistent with fibroblast-like cells (*A2M*, *PTGDS*), myofibroblasts (*ACTA2*, *RGS5*) were reproducibly detected in AAH and AIS tumors, so they may be features of the TME in early-stage LUAD (Fig. S16b). Fibroblast-like cells and myofibroblasts positive for α-smooth muscle actin (α-SMA), encoded by the *ACTA2* gene, function as cancer-associated fibroblasts (CAFs), promoting extensive tissue angiogenesis[45] and tumor progression[46]. Smooth muscle cells were observed in IA stage tumors and a few normal tissues. These cells are the main type of fibroblasts in vasculature and have been linked to wound healing and angiogenesis[47]. GSEA comparing fibroblasts from normal and tumor tissues showed that cancer-derived fibroblasts were associated with the oxidative phosphorylation and with strong IFN-γ and IFN-α responses[15] (Fig. 4f). These may be related with the increased synthesis and secretion of collagens[48]. Our results suggest that stromal cells shift towards a phenotype of tissue remodeling and angiogenesis during LUAD progression.

Lymphocytes play important roles in inflammation, cancer immune evasion, and responses to immunotherapy treatment[49]. Our dataset of 61,196 lymphocytes consists of 10 clusters, mainly T cells, B cells and NK cells, among other immune cell types

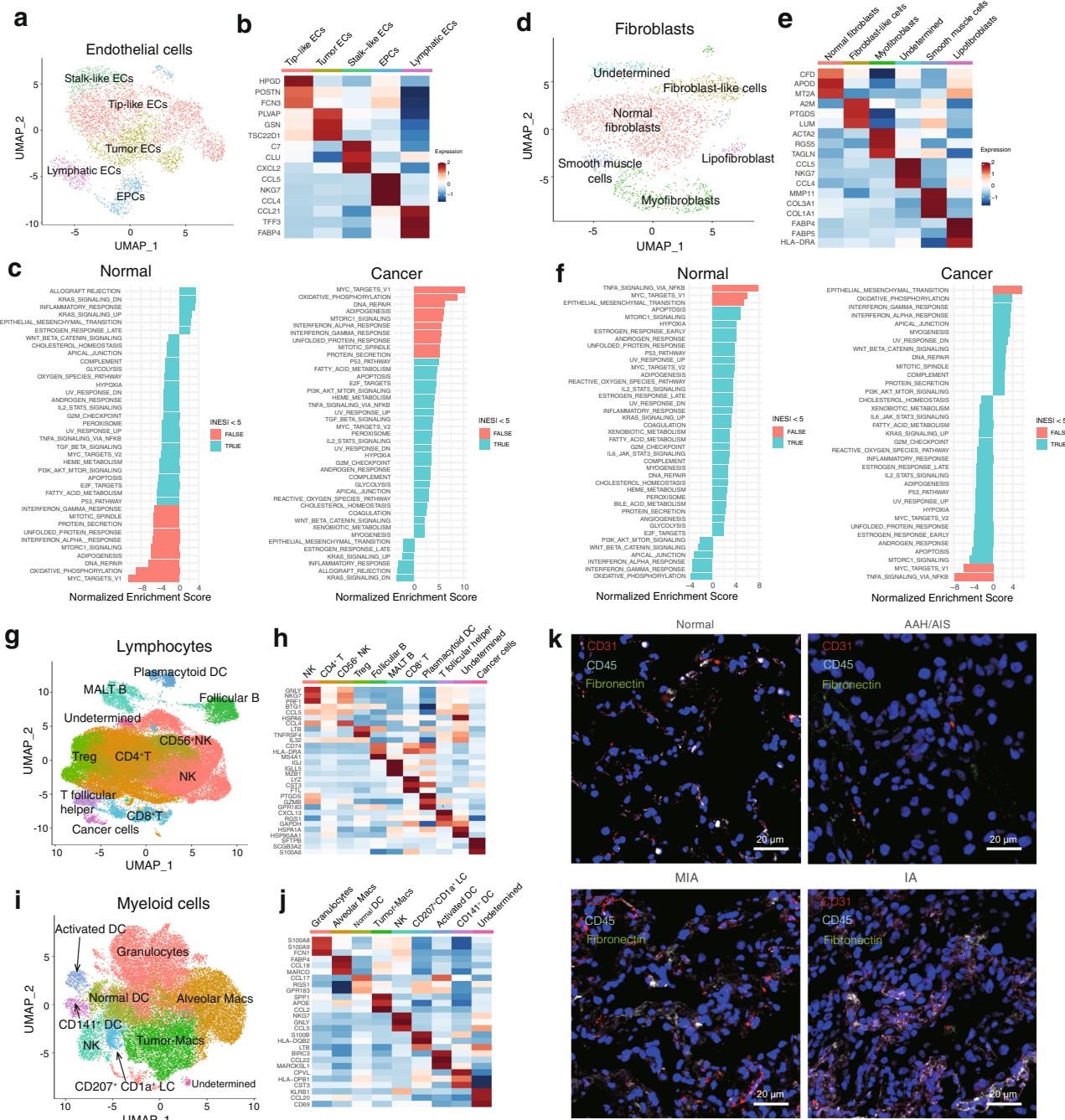

**Fig. 4 Profiling stromal and immune cell populations in LUAD progression. a** UMAP visualization of endothelial cell clusters, color-coded by identified cell subtypes. ECs: endothelial cells; EPCs: endothelial progenitor cells. **b** Heatmap of selected marker genes in EC cluster subtypes. **c** Top enriched pathway of ECs isolated from normal or tumor tissues as determined by GSEA. **d** UMAP visualization of fibroblast clusters, color-coded by identified cell subtypes. **e** Heatmap of selected marker genes in the fibroblast cluster subtypes. **f** Top enriched pathway of fibroblasts isolated from normal or tumor tissues as determined by GSEA. **g** UMAP visualization of lymphocyte clusters, color-coded by identified cell subtypes. **h** Heatmap of selected marker genes in the lymphocyte cluster subtypes. **i** UMAP visualization of myeloid cell clusters, color-coded by identified cell subtypes. **j** Heatmap of selected marker genes in the myeloid cell cluster subtypes. **k** Co-immunofluorescence staining for ECs (CD31, red), immune cells (CD45, gray) and fibroblasts (fibronectin, green) in normal and tumor tissues from different stages. Nuclei were stained blue (DAPI). Each staining in panels come from three samples. Scale bars: 20 μm. GSEA: Gene Set Enrichment Analysis; UMAP: Uniform Manifold Approximation and Projection.

(Fig. 4g, h and Fig. S15). We observed that CD8+ T and regulatory T (Treg) cells were enriched in the tumor, but that natural killer cells were depleted during tumor progression (Fig. S4c, d and Fig. S15c). We found HLA-DRA, as an exhaustion marker, expressed in CD8+ T cells[50]. IL32 showed

higher expression in CD4+ T and Treg cells, which was previously reported to be associated with immune suppression[51]. This is consistent with the idea that T cell-mediated cytotoxicity is critical for tumor cell clearance[52]. While B cells and plasma cells were rare in most samples. Our analysis suggested a shift in

lymphocyte composition and gene expression towards immune suppression during LUAD progression.

Myeloid cells play a critical role in maintaining tissue homeostasis, and they regulate inflammation in the lung[19]. We examined the gene signatures of the 8 myeloid clusters including granulocytes, macrophages, NK cells and dendritic cells (Fig. 4i, j and Fig. S15). Dendritic cells were mostly myeloid cells falling into three DC subsets: DCs (*CCL17*), activated DCs (*BIRC3*, *CCL22*) and CD141+ DCs (*CPVL*, *CST3*). CD141+DCs, which express lymphotoxin beta transcripts in lung tumor tissues and contribute to tertiary lymphoid structure formation[53], were significantly less abundant in LUAD tissues than in normal lung. Alveolar macrophages (AMs), which highly express *MARCO*, *FABP4*, and *MCEMP1*[37], were detected mainly in normal and early-stage LUAD. Tumor macrophages (TMs) comprised the remaining tumor-enriched clusters and were present mostly in IA tumors (Fig. 4i, Fig. S4d, and Fig. S15a). TMs showed high expression of *SPP1*, *APOE* and *CCL2*, involved in apolipoprotein metabolism[37]. We speculate that the TMs in the IA stage induce tumor angiogenesis, promote tumor migration, invasion, and form an immunosuppressive TME[54].

TME is heterogeneous and includes reprogrammed immune cells, fibroblasts, and ECs[14]. To characterize stromal cell heterogeneity in LUAD progression, we performed simultaneous immunofluorescence staining for ECs (CD31), fibroblasts (fibronectin) and immune cells (CD45) in normal and tumor tissues from patients of different stages (Fig. 4k and Fig. S17). The CD45+ and Fibronectin+ populations increased as the tumor progressed, consistent with the idea that immune cell infiltration and EMT help define the TME during tumor progression[55]. In fact, we found that the levels of six cell subtypes, including CD4+ T cells, T follicular helper cells, activated DC cells, and granulocytes have a negative association, while other cell subtypes for plasmacytoid DC cells and CD141+ DC cells, have a positive association with the survival period (Fig. S18). These findings suggest that stromal cells and immune cells in the lung TME may predict clinical outcome.

**Cell–cell crosstalk during LUAD progression**. The hematopoietic stromal cell lineage and tumor epithelial cells appear to engage in cell-type-specific crosstalk[56]. We used CellPhoneDB to identify the expression of potential crosstalk signaling molecules based on ligand-receptor interactions[57]. The epithelial cell clusters, especially the AT2/AT2-like clusters, as well as the fibroblast clusters showed the most interactions with other cell types, such with myeloid cells (Fig. 5a and Fig. S19). This suggests interactions between epithelial and stromal cells involving certain receptor-ligand gene pairs (Fig. 5b). We focused on gene pairs associated with $p < 0.001$ in strongly interacting cell types. AT2-like cells express high levels of ANXA1, MDK, and FN1 (Fig. 5c), the receptor of FPR1, SORL and a4b1 were expressed by DCs and macrophages. The results were consistent with our comparison of expression patterns between normal tissue and tumors in different histologic stages (Figs. S20 and S21). The expression pattern of the FN1-A4B1 (A4B7) and ANXA1-FPR1(FPR3) receptor-ligand complex indicates the existence of functional interactions between AT2-like cells and immune cells. By contrast, AT2 cells expressed higher levels of LGALS9, as a major binding protein for PD-1 with known immunomodulatory activity[58], the ligands of COLEC12 and MRC2 were found in DCs, granulocytes and macrophages. Endothelial cells and fibroblasts strongly expressed receptors such as FN1 and CXCL12, which can interact with immune-related ligands. For example, fibroblasts express higher levels of CXCL12, the receptor of CXCR4, which is widely expressed on immune cells. These cytokines have been associated with metastasis of cancer[49].

To characterize potential signaling crosstalk between immune cells and epithelial cells, normal and tumor tissues were stained for simultaneous detection of cytokeratin (tumor cells), CD8 (cytotoxic T cells), FoxP3 (regulatory T cells), CD68 (macrophages), PD-1, and PD-L1 (Table S1). The results confirmed that infiltration of immune cells increased with LUAD progression (Fig. 5d and Fig. S22), and existing in a high spatiotemporal intertumoral heterogeneity in IA stage. However, we did not detect clear interactions between tumors and immune cells. Statistical tests for fluorescence intensity were performed between pan-CK+ vs. pan-CK− cells from normal to AAH/AIS, MIA, and IA (Fig. S23). The results shown that the PD-1 was high expression for pan-CK− cells in AAH/AIA stage, while inversely PD-1 was high expression for pan-CK+ cells in MIA an IA stages. This is in line with a similar study showing that PD-1 could be expressed in tumor cells and could activate mTOR or Hippo signaling pathway, therefore facilitating tumor proliferation[59]. Taken together, interactions related to immunomodulatory signaling were more abundant in LUAD in comparison with normal, indicating heterogeneity and plasticity of the tumor ecosystem, varying during cancer progression.

## Discussion

TME is composed of multiple cell types[14], while the cells within a tumor can show a substantial level of heterogeneity[60]. Here we provide a high-resolution scRNA-seq dataset of LUAD cells collected from four different histologic stages to recapitulate key transcriptional events during LUAD progression. Our findings provide valuable insights into the pathogenesis of LUAD early cancer progression, including AT2 cells being the most likely cancer cells of origin. In addition, we also discovered that tumor ECs are highly angiogenic, yet immune compromised. Fibroblast-like cells and myofibroblasts are CAFs that promote tumor progression. CD8+ T cells and Tregs persist in the IA stage, providing a suppressive mechanism antitumor immunity during tumor progression. Transcriptional phenotypes in TMs, which are involved in apolipoprotein metabolism, are observed mostly in IA stage. Consistent with recent findings, these alterations in stromal and immune populations cooperatively transformed immune-competent tissues into an immune-suppressive TME during LUAD progression[18,48,61]. Eventually, we find that *MDK* and *TIMP1* are potential biomarkers to facilitate our understanding of LUAD pathogenesis.

The heterogeneity of tumor cells represents a major challenge in oncology. We found that the level of transcriptional heterogeneity dramatically increased during LUAD progression. Different epithelial cell types in our samples exhibited unique molecular signatures. In fact, AT2 cells and other alveolar progenitor cells have been reported to participate in the repair of alveoli[62]. In alveoli, AT2 cells self-renew under homeostatic conditions and initiate the generation of stem-like cells after injury or gene mutation[6]. Early cancer cells dedifferentiated into a stem-like state that closely resembled AT2 cells, which we termed AT2-like cells, and gave rise to the heterogeneous populations of cancer cells observed in LUAD (Fig. 6). Our work suggested that LUAD cancer progression could be initiated by progressive downregulation of tissue-specific marker genes such as *SFTPC*, and upregulation of stem cell signaling factors such as *CD44*. AT2-like cells strongly express ribosomal and mitochondrial genes that promote tumor progression. This phenotype was similar to a recently reported pulmonary subsolid nodules showing a strong metabolic reprogram[61]. Thus, an increase in the expression of ribosomal and mitochondrial genes could be an early indicator of lung cancer. We also observed that Lgr5+ LUAD cells display persistent proliferative potential, followed by

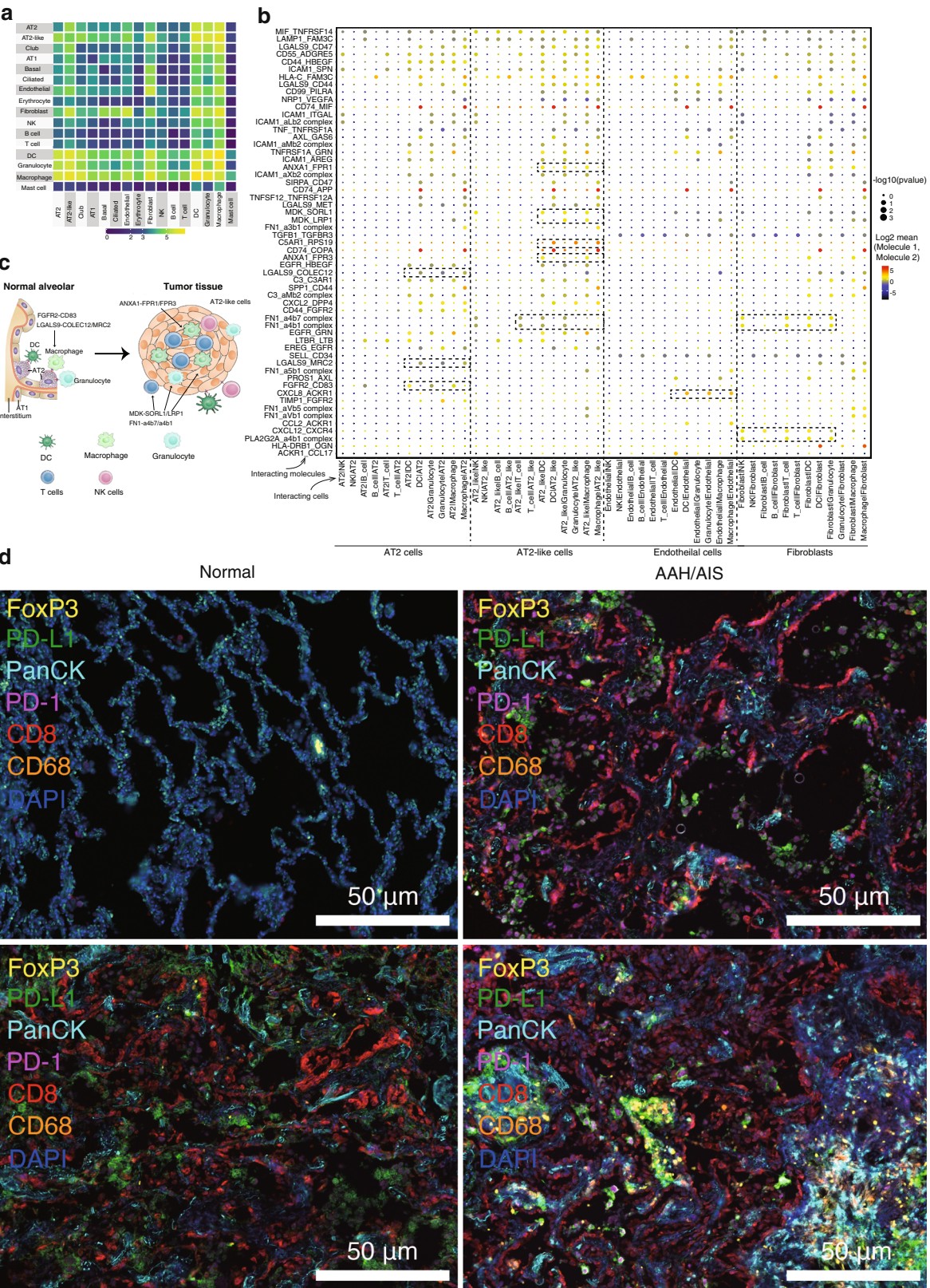

the appearance of several alternative differentiation programs mimicking the primordial lung, and then the emergence of cells with a mesenchymal phenotype, indicating the completion of the EMT. Throughout this process, we observed the expression of EMT, migration and stemness genes such as *E-cadherin*, *IFI27* and *S100A4*[30,63–65]. Taken together, our finding shed light on the

underlying transcriptional signatures of distinct histologic stages of LUAD progression.

Immune cells can communicate via ligand-receptor interactions[56], so targeting cell–cell interactions is frequently utilized in the clinical setting. For example, the immune check-point inhibitor ipilimumab targets the binding of ligands to CD28

**Fig. 5 Cell–cell crosstalk during LUAD progression. a** Heat map depicting the significant interactions among the 16 major cell types identified in Fig. 1b. **b** Overview of the selected ligand-receptor interactions. *p*-values (two-tailed permutation test) are indicated by circle size; the scale is on the right. The means of the average expression level of interacting molecule 1 in cluster 1 and interacting molecule 2 in cluster 2 are indicated by color. Assays were carried out at the RNA level, but extrapolated to protein interactions. Selected cells include AT2 cells, AT2-like cells, ECs and fibroblasts. AT2: alveolar type 2 cells; ECs: endothelial cells. **c** Diagram of the main receptors and ligands expressed on AT2 cells and AT2-like cells, AT1: alveolar type 1 cells; AT2: alveolar type 2 cells. **d** Representative multiplexed staining of cytokeratin-positive tumor cells (Cyan), CD68+ macrophages (orange), FoxP3+ regulatory T cells (yellow), CD8+ T cells (red), PD-1+ cells (magenta), and PD-L1+ cells (green) on tissues from different stages. Each staining in panels come from three samples. Scale 50 μm, nuclei (DAPI) are stained blue.

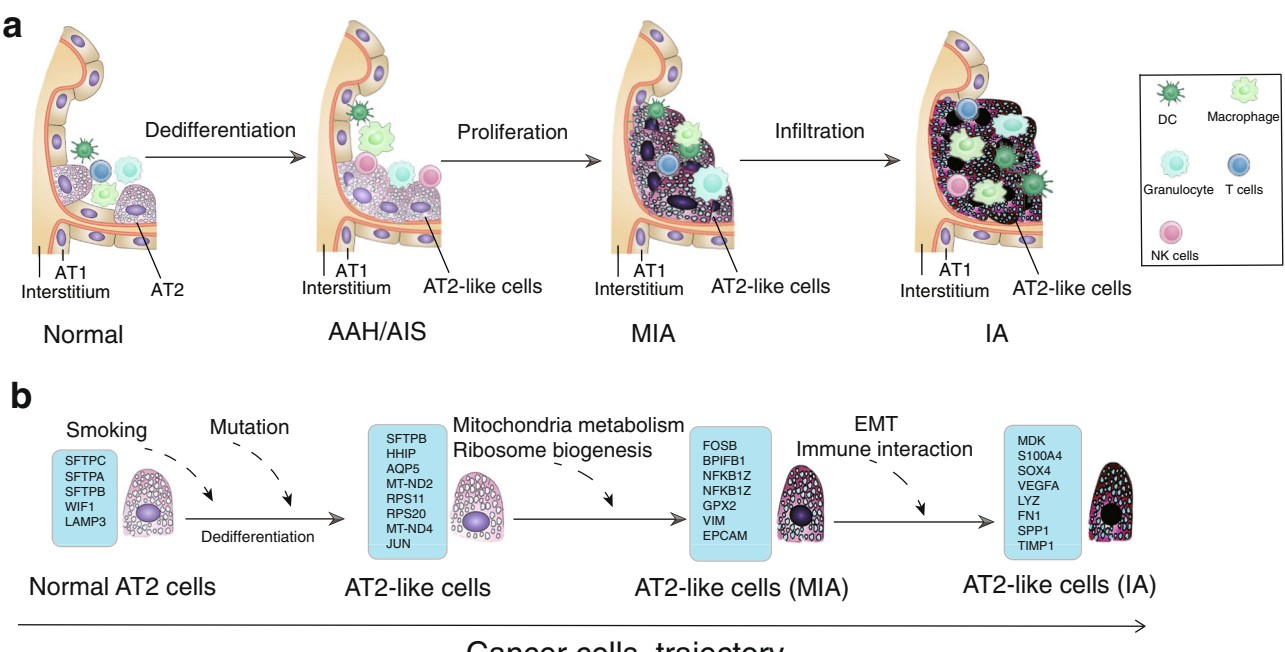

**Fig. 6 Schematic overview of the cellular and molecular mechanisms involved in the cancer progression of LUAD.** The proposed cellular (**a**) and molecular (**b**) mechanisms in cancer cells trajectory. AT1: alveolar type 1 cells; AT2: alveolar type 2 cells; AAH: atypical adenomatous hyperplasia; AIS: adenocarcinoma in situ; MIA: minimally invasive adenocarcinoma; IA: invasive adenocarcinoma; EMT: epithelial-mesenchymal transition.

or CTLA4, and both pembrolizumab and nivolumab target the interaction between PD1 and PD-L1[66]. Malignant cells activate cellular and humoral immune responses to different extents in different patients, while also inducing immune exhaustion. AT2-like cells express high levels of ANXA1, MDK and FN1, which bind receptors of FPR1, SORL and a4b1, were expressed by DCs and macrophages in IA stage. However, AT2 cells interact specifically with myeloid cells through the LGALS9 receptor. Intercellular interactions suggest that AT2-like cells might be responsible for the recruitment of DCs and macrophages by specific ligands. Overall, our single-cell data may help explain tumor-immune interactions by providing insights into the composition and dynamics of T cells in LUAD progression. However, our knowledge of which interactions occur in tumors and how these interactions affect prognosis are still limited. There is a need to elucidate the full spectrum of cell–cell interactions in the TME and how these interactions affect patient outcomes.

One limitation of our study was the sample representation. In the clinical setting, the primary tumor tissues from the same patient along the trajectory of tumor progression could only be collected once, serial sample collection was not possible to do a longitudinal study. However, we were able to collect multiple tumor tissues at different histologic stages from the same patient, which provide valuable insights into the study on early LUAD carcinogenesis. Secondly, it is very difficult to obtain thousands of cells per patient with LUAD in early stages such as AAH and AIS in the clinical setting due to the size of the lesions. Nonetheless, Tirosh et al. have previously identified distinct tumor microenvironmental patterns using just few thousand single metastatic melanoma tumors isolated from a total of 19 patients[67]. So single-cell data obtained from small sample size could still be meaningful. Third, it is worth noting that the tissue samples used for single-cell sequencing and pathological diagnosis were different parts of the same surgical specimens, so there might be potential differences in their biological complexities of these parts. Finally, our findings could be further validated and extended with GEMM and PDX approaches, preferably with integration of genomics, transcriptomics and proteomics to capture lineage plasticity comprehensively. Beyond the outlined technical challenges, trajectory inference analysis also has its limitation. The field is still maturing, and the complexity of the underlying topology could be underestimated[68]. Therefore, future studies on larger cohorts and utilizing tools designed to handle increasingly complex biological features are needed to dissect the evolutionary trajectory of early LUAD carcinogenesis and its underlying molecular mechanisms.

In summary, we constructed a single-cell transcriptome atlas of premalignant lung lesions and LUAD at all major clinical stages, which highlights the transcriptional heterogeneity of lung epithelial cells at different tumor stages. We found that each epithelial cell-type exists in a unique transcriptional state, but that

early cancer cells are highly similar to AT2 cells, which undergo dedifferentiation to generate a stem-like state that initiates and maintains tumor progression. These stem-like cells give rise to the heterogeneous populations observed in LUAD. In addition, our research has shown that *MDK* and *TIMP1* are potential biomarkers to facilitate understanding of LUAD pathogenesis.

## Methods

**Patients and clinical information**. This study was designed to characterize the cellular heterogeneity and molecular events underlying LUAD initiation and progression. We performed single-cell RNA sequencing on 52 specimens from 25 patients spanning the four histologic stages (AAH, AIS, MIA and IA), together with 18 of matched normal lung samples. Patients diagnosed with AAH, AIS, MIA, and IA according to the 2015 WHO classification[69], were enrolled with informed consent from West China Hospital of Sichuan University, China. All patients received surgical treatment and none of them underwent neoadjuvant therapy before surgery. Tumors and matched distal normal lung tissues were obtained during surgery. Cancer clinical stage was defined according to the 8th edition of the American Joint Committee on Cancer (AJCC) TNM stage system. This study protocol was approved by the Institutional Review Board of West China Hospital of Sichuan University (Ethics: Project identification code: 2018.270). All the patients have provided written informed consent.

**Sample preparation**. Resected tumors were transported in Hank's Balanced Salt Solution (HBSS, Life Technologies) on ice immediately after surgery. The tumor sample was subsequently divided into two pieces, and a small fragment was stored in liquid nitrogen for tissue staining. The remainder of the tumor was minced with scalpels into tiny cubes <0.5 mm³ and transferred into a 15 mL conical tube (BD Falcon) containing 8 mL pre-warmed HBSS, 1 mg/mL collagenase I and 0.5 mg/mL collagenase IV. Tumor pieces were digested on a Tube Revolver (Thermo) for 30 min at 37 °C. This suspension was then filtered using a 70 μm nylon mesh (BD Biosciences) and residual cell clumps were discarded, then the cell pellet was resuspended in red blood cell lysis buffer. Following a 5 min incubation at room temperature, samples were centrifuged to discard the supernatant and re-suspend the cell pellet in PBS with 0.04% FBS. Cell sorting was performed with a MoFloAstrios EQ (Beckman Coulter). Live cells were used for single-cell experiments after the dead cells were eliminated based on exclusion of 7-aminoactinomycin D (Life Technologies).

**Genome sequencing**. DNA was extracted from freshly frozen tumors and matched normal samples using the Gentra Puregene DNA Extraction Kit (Qiagen) following the protocol of the manufacturer. WES was performed with hybridization-captured; adapter ligation-based libraries were synthesized using an Agilent SureSelect Human All Exon Kit (Santa Clara, CA, USA), which was designed to enrich 334,378 targeted exonic regions of 20,965 genes. WES libraries were then amplified, quality-checked, and sequenced using an Illumina NovaSeq 6000 Platform. The average sequencing depth of the target regions was >200×. For WGS, library was generated using Truseq Nano DNA HT Sample Prep Kit (Illumina USA) following the manufacturer's recommendations, and index codes were added to each sample, then sequenced on Illumina NovaSeq 6000 Platform, the average sequencing depth was 30×.

**Genome data processing**. Pair-end reads were aligned to hg38 reference genome via BWA-mem (version 0.7.13-r1126), we use SAMtools (version 1.9) to sort and index BAM and Picard Tools (version 2.2.1) to mark duplicates. The Genome Analysis Toolkit (GATK, version 4.1.2.0) was used to local realignment and base quality recalibration. Germline single-nucleotide variants (SNVs) and indels were detected by GATK HaplotyeCaller, somatic SNVs and indels were detected by GATK mutect2, both with default parameters. We use ANNOVAR (version 2020.06.07) to facilitate the variant annotations with parameter -dbtype wgEncodeGencodeBasicV33. Variants which located in the exon of 9 genes (*BRAF*, *EGFR*, *ERBB2*, *HRAS*, *KRAS*, *MAP2K1*, *MET*, *NF1* and *ROS1*) we interested were selected basing on the results of ANNOVAR.

**scRNA-seq library preparation and sequencing**. Single-cell suspensions were converted to barcoded scRNA-seq libraries using the Chromium Single Cell 3'Library, Gel Bead & Multiplex Kit and Chip Kit (10x Genomics) following the manufacturer's instructions. The goal was to achieve approximately 5,000 cells per library. The sequencing-ready library was purified with SPRIselect, quality-controlled for size distribution and yield (LabChip GX Perkin-Elmer) and quantified using quantitative PCR (KAPA). Libraries were sequenced on an Illumina NovaSeq-6000 system and mapped to the human genome (build hg19) using CellRanger (10x Genomics).

**Single-cell RNA sequencing analysis and identification of marker genes**. Raw gene expression matrices generated per sample using CellRanger (version 3.0.0) were combined in R (version 3.6.3) and converted to a Seurat object using the Seurat R package (version 3.0.3.9028). Cells were removed if they had more than 20,000 UMIs, more than 3,000 or fewer than 300 expressed genes, or >10% UMIs that were derived from the mitochondrial genome. After filtering, the gene expression matrices were normalized to total cellular read count, original sample identity, and mitochondrial read count using linear regression as implemented in Seurat's "Regress Out" function. Consequently, none of the resulting principle components correlated with transcript count.

To reduce dimensionality of this dataset, the variably expressed genes were summarized by principle component analysis, with the first 100 principle components further summarized using UMAP dimensionality reduction with the default settings in the RunUMAP function. Clustering was conducted using the "FindClusters" function using 50 PCA components with resolution parameter set to 2. Cell clusters in the resulting two-dimensional representation were annotated to known biological cell types using canonical marker genes. Very few cells were positive for cell proliferation markers, so we did not correct for effects of cell cycle in the analysis.

To identify marker genes of cell clusters, we contrasted cells from one particular cluster to those in all other clusters using the Seurat "FindAllMarkers" function. Marker genes were required to have an average expression in one particular cluster that was >2.5-fold higher than that in the other clusters.

**Gene set enrichment analysis**. GSEA is a widely used approach to test whether a particular gene set is enriched at the top of a ranked gene list[70]. The fgsea package (version 1.8.0) was used with default settings together with annotated Hallmark gene sets from the msigdbr package (version 7.2.1). The top 50 pathways ranked by adjusted *p*-value (adjusted *p* < 0.005) were plotted in the visualization.

**Trajectory analysis**. In order to generate a trajectory, we generated a randomly sampled subset of malignant cells from each histologic stage among the epithelial cells in the samples of lung tumor tissue. Next, we employed the Monocle2 (version 2.12.0) algorithm using the gene-cell matrix in the scale of UMI counts extracted from Seurat subset as input, and a new "Cell Data Set" function was used to create an object with the parameter negbinomial size as the expression family. The cell trajectory was inferred using default parameters after dimension reduction and cell ordering.

**InferCNV and clonality analysis**. For the InferCNV (version 1.0.4) analysis, the following parameters were used: "denoise", default hidden Markov model (HMM) settings, and a "cutoff" of 0.1. To reduce the possibility of false positives, CNV calling of the default Bayesian latent mixture model was implemented to identify the posterior probabilities of alterations in each cell. Low-probability CNVs were filtered using the default threshold of 0.5. To determine the clonal CNV changes in each tumor, the "subcluster" method was utilized on the CNVs generated by the HMM. GRCh37(hg19) cytoband information was used to convert each CNV to a p- or q- arm-level change for simplification based on its location. Each CNV was annotated to be either a gain or a loss. After data conversion, subclones containing identical arm-level CNVs were collapsed. Chromosomes X and Y as well as mitochondrial CNVs were excluded from this analysis. UPhyloplot2 (version 2.3) was used to generate evolutionary trees with default parameters. A scalable vector graphics (.svg) file visualizing the phylogenetic tree was generated for each sample.

**Gene regulatory network analysis**. We constructed the gene regulatory network in normal lung cells and cancer cells using the bigSCale2 (version 2.0) algorithm (https://github.com/iaconogi/bigSCale2). Briefly, the expression data of the 13,461 cells in the AT2 and AT2-like cell clusters were extracted using Seurat and then combined into a sparse expression data matrix. We eliminated the genes expressed in fewer than 10% of cells, leaving us with 3,613 genes for the network analysis. The resulting matrix was then input into bigSCale2 for construction of the network under the "direct" clustering parameter; only genetic interactions with the correlation coefficient > 0.75 were retained. The network was then visualized using the Prefuse Force Directed Layout in Cytoscape (version 3.8.0) (https://cytoscape.org/).

**Cell–cell interaction network analysis**. We mapped cell–cell interaction and receptor-ligand pairs between all major cell types using CellPhoneDB (version 2.1.2) (www.cellphonedb.org). Potential interactions between the two cell types were inferred through gene expression levels through 1000 permutation tests. Then the resulting adjacency matrices were generated for all cell–cell interactions and visualized on heatmaps. Cell–cell interactions within identical cellular lineages were excluded, and only gene pairs for receptor-ligand interactions in cell types of interest were visualized, so long as the combined *p*-value < 0.001 (obtained by multiplying all *p*-values within each gene-pair).

**SCENIC analysis**. R package SCENIC (version 1.1.2) to infer gene regulatory network activity, in which we scored the activity of each regulon of the single cells using the default settings and the following cisTarget databases: hg19-500bp-upstream-7species.mc9nr.feather (https://resources.aertslab.org/cistarget/databases/homo_sapiens/hg19/refseq_r45/mc9nr/gene_based) and hg19-tss-

centered-10kb-7species.mc9nr.feather (https://resources.aertslab.org/cistarget/databases/homo_sapiens/hg19/refseq_r45/mc9nr/gene_based).

**Survival analysis based on the Cancer Genome Atlas**. To assess the correlation of specific cell types with survival of LUAD patients, we downloaded LUAD RNA-seq data (TCGA-LUAD), as well as clinical data using the Bioconductor TCGA biolinks package (version 2.2.10). 515 LUAD RNA-seq data (TCGA-LUAD) as well as clinical data were used. For the cell cluster identified in the study, the top 10 marker genes in each cluster were ranked by log (fold change), and averaged per patient and assigned to high or low groups based on the median expression. Survival analyses were performed using the R packages "survival" (version 3.2.3) and "survminer" (version 0.4.7). Kaplan–Meier survival curves were generated for high and low groups, then compared using Cox regression $p$-values after correction for age, sex, and tumor stage.

**Immunofluorescence staining**. Human lung normal and tumor tissues were fixed by perfusion in 10% paraformaldehyde (PFA) for 48 h at 4 °C. Formalin-fixed, paraffin-embedded sections (4 μm) were deparaffinized in xylene and hydrated in graded alcohol. The dried sections were washed three times with phosphate buffer saline (PBS), then blocked with normal goat serum. Sections were stained with primary antibodies at 4 °C overnight in a wet box containing a small amount of water, then washed with PBS three times, and finally incubated with Alexa-conjugated secondary antibodies for 50 min at room temperature. Sections were then washed with PBS three times and mounted with mounting medium containing DAPI (Vector Labs).

Primary antibodies against the following proteins were used: SFTPC (Millipore, AB3786, 1:200), SFTPA (Millipore, AB3420-I, 1:200), AGER (R&D, AF1145, 1:50), SCGB1A1 (Santa Cruz, SC-365992, 1:100), FOXJ1(Abcam, ab235445, 1:200), Nkx2-1(Millipore, SAB1403709, 1:500), TIMP1 (Invitrogen, MA5-13688, 1:200), EPCAM (Abcam, ab223582, 1:50). CD45 (Servicebio, GB14038, 1:100), CD31(Servicebio, GB14033, 1:200), VEGF (Servicebio, GB14165, 1:200), Vimentin (Servicebio, GB111308, 1:1500), Fibronectin (Servicebio, GB13091, 1:100), Ki67 (Servicebio, GB14102, 1:200), and MDK (Abcam, ab215835, 1:50). The secondary antibodies were Alexa Fluor 488-conjugated AffiniPure Donkey Anti-Rabbit IgG (H + L) (min X Bovine, Chicken, Goat, Guinea Pig, Syrian Hamster, Horse, Human, Mouse, Rat, Sheep Serum Proteins, Jackson, 711-545-15, 1:500); Cy3-conjugated AffiniPure Donkey Anti-Mouse IgG (H + L) (min X Bovine, Chicken, Goat, Guinea Pig, Syrian Hamster, Horse, Human, Rabbit, Rat, Sheep Serum Proteins, Jackson, 715-165-15, 1:500); Alexa Fluor 647-conjugated AffiniPure Donkey Anti-Goat IgG (H + L) (min X Chicken, Guinea Pig, Syrian Hamster, Horse, Human, Mouse, Rabbit, Rat Serum Proteins, Jackson, 705-605-14, 1:500). Pictures were taken with a Zeiss fluorescence microscope (Imager. Z2) system.

**Multiplex immunohistochemistry (OPAL™) staining**. Human lung normal and tumor tissues were fixed in 4% FPA and embedded in either paraffin or OCT (Tissue-Tek O.C.T., Sakura Finetek, USA). Formalin-fixed, paraffin-embedded samples were sliced into 4 μm thick sections. Consecutive staining was performed by heat-induced antigen retrieval followed by incubation with primary antibody with the Opal Polaris 7-Color Manual IHC Kit (NEL861001KT). The panel kit MOTiF™ PD-1/PD-L1 (OP-000001, Akoya, USA) was used to simultaneously label regulatory T cells (FoxP3), tumor cells (Pan CK), cytotoxic T cells (CD8), tumor-associated macrophages (CD68), and immune check point markers (PD-1/PD-L1) on the same tissue slide. The chromogen-based multiplex immunohistochemistry labeling was operated by an automated staining system (BOND-RX; Leica Microsystems, Vista, CA). Antibodies in the kit were in working dilution, including Pan CK (clone AE1/AE3), FoxP3 (clone D608R), PD-L1 (clone E1L3N), PD-1 (clone, EPR4877), CD8 (clone 4B11), and CD68 (clone PG-M1). The Opal Polaris dyes were paired with a panel of antibodies containing fluorophores for tyramide signal amplification (TSA) to enhance sensitivity. The sequence of labeling for detecting each marker was optimized: Foxp3 (Opal 570), PD-L1 (Opal 520), Pan CK (Opal 690), PD-1 (Opal 620), CD8 (Opal 480), CD68 (Opal 780), and DAPI. One autofluorescence section was included, which was stained using the same procedure as above but without adding primary antibody. Multiplex analysis was performed to analyze the simultaneously stained samples.

**Image and analysis**. Stained sections were imaged using a Vectra Polaris automated quantitative pathology imaging system (Akoya, USA) at high resolution (20x). The images were spectrally unmixed by Phenoptics inForm software (inForm 2.4.8, Akoya, USA). Subsequent cellular and subcellular analyses were performed using inForm software. Within tumor tissues, Pan CK positivity was used to define the tumor area, while stroma was defined as Pan CKnegative. Using inForm software, we quantified the area and percentage of each tissue type. In each tissue region, cells were recognized and separated according to the DAPI signal. Furthermore, individual fluorophore signal intensity was calculated in each cell. The positivity of each channel was quantified using manually defined thresholds of positivity.

**In situ RNA-FISH**. Formalin-fixed, paraffin-embedded biopsies were sectioned to generate 5 μm thick sections. All materials, including the microtome and blade, were sprayed with RNase-away solution prior to use. Slides were baked for 1 h in a 60 °C dry oven the night before, and stored overnight at room temperature in a Parafilm-sealed slide box containing a silicone desiccator packet. Sections were deparaffinized and dried at room temperature. Protease activity was blocked and the slides were incubated with probes (Lgr5: TCCCCAAAAGGCAAAGGCAGG-CAGAGAG, E-cadherin: TGGTGTAAG- CGATGGCGGCATTGTAGGTGT, FOXM1: GATCTTGCTGAGGCTGTCATTC-ATTGTG, Vimentin: CAA-GACGTGCCAGAGACGCATTGTCAACAT and EPCAM: CCAACTGAAGTA-CACTGGCATTGACGATT). The fluorochromes Cy3(Lgr5) and DIG (E-cadherin, FOXM1, Vimentin and EPCAM) were bound to the probes, and the nuclei were counterstained with DAPI. The slides were covered with a coverslip and imaged under a Nikon Imaging system (NIKON DS-U3) and Caseviewer software (version 2.3). Imaging parameters were kept consistent for all images within the same experiment, and any post-imaging manipulations were performed in the same way on all images from a single experiment.

**Statistics and reproducibility**. No statistical method was used to predetermine sample sizes. Samples were processed for scRNA-seq (10x Genomics) soon after resection in the operating room. As a result, samples from different patients were processed in separate experiments. Survival probabilities were estimated using the Kaplan–Meier method, and differences between Kaplan–Meier curves were compared using the log-rank test. Univariable and multivariable Cox proportional hazard regression models were used to identify independent prognostic factors. Hazard ratios (HR) were presented together with their 95% confidence intervals (CI) and corresponding $p$-values. Statistical tests were two-sided, and a $p$-value of <0.05 was considered significant (*$p < 0.05$, **$p < 0.01$, ***$p < 0.001$, ****$p < 0.0001$). Non-parametric Kruskal–Wallis tests were applied as described in the corresponding figure legends. Analyses were performed using SPSS software (version 18.0, SPSS Inc., Chicago, IL, USA). A Spearman correlation matrix was generated to examine associations among different cell types. All representative images reflect a minimum of three biological replicates.

**Reporting summary**. Further information on research design is available in the Nature Research Reporting Summary linked to this article.

## Data availability
The generated WES, WGS, and RNA-seq data in this study have been deposited to Genome Sequence Archive (GSA) in BIG Data Center, Beijing Institute of Genomics (BIG) under accession number HRA001130. The transcriptome data of TCGA LUAD were collected from the following web-links https://portal.gdc.cancer.gov/projects/TCGA-LUAD. The human-specific databases for RcisTarget were downloaded from (https://resources.aertslab.org/cistarget/databases/homo_sapiens/hg19/refseq_r45/mc9nr/gene_based/hg19-500bp-upstream-7species.mc9nr.feather) and (https://resources.aertslab.org/cistarget/databases/homo_sapiens/hg19/refseq_r45/mc9nr/gene_based/hg19-tsscentered-10kb-7species.mc9nr.feather). Source data are provided with this paper.

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

## Acknowledgements

We are thankful to all the patients for their voluntary participation in the study. We thank Dr. Kun Zhang from the University of California San Diego, and Wan Xiong from West China Hospital for valuable discussions. This work was supported by the National Natural Science Foundation of China (81871890, 91859203 to W. Li; and 81802300, 32170592 to Z. Wang), CAMS Innovation Fund for Medical Science (2019TX310002 to W. Li), the China Postdoctoral Fund (2019T120850 to Z. Wang), the Sichuan Science and Technology Program (2019YJ0159 to Z. Wang; and 2018SZ0229 to H. Xue), the Chengdu Science and Technology Program (2019YF0500373SN to G. Che), and the

National Guided Science and Technology Development Project of Sichuan Province (2020ZYD009 to W. Li).

## Author contributions

Z.W., W.L., and W.Q. performed single-cell RNA sequencing and staining experiments. G.W. and Y.N. performed sequencing experiments and processed the data. Z.L., M.X., Y.L. and Z.S. performed bioinformatic analyses. K.Z., C.W. and G.J. obtained patient consent and collected the samples. S.D. assisted in participant selection, consent, clinical information and procurement of tissue. L.J. and L.Z. performed histological evaluations. Z.W., Y.Y., and T.G. performed fresh tissue dissociations, immunostaining, microscopy and imaging. W.L. and G.C. provided clinical insights. Z.W. and Z.L. analyzed and interpreted the data. Z.W. and Z.L. conceived the study and wrote the manuscript. All authors reviewed and edited the manuscript.

## Competing interests

The authors declare no competing interests.
