## [Peer review file · Nature Communications]

REVIEWER COMMENTS

Reviewer #1 (Remarks to the Author): Expert in single-cell RNA-seq and lung cancer genomics

Review comment

In this study, Wang, Li, Zhou, and colleagues applied single-cell RNA sequencing to precancerous lesions of lung adenocarcinoma and characterized stepwise alterations from the non-malignant lung towards atypical adenomatous hyperplasia (AAH), adenocarcinoma in situ (AIS), minimally invasive adenocarcinoma (MIA), and invasive adenocarcinoma (IA). As the authors claimed, the study provides a valuable resource for the understanding of early events during the development of lung adenocarcinoma. However, most of the results are presented without statistical tests and sometimes flawed by interpretation of trajectory analysis on multi-lineage cells (cancer and immune cells) or potential doublets or auto-fluorescent cells. Over-interpretation and statements without basis afflict the entire manuscript.

Major comments

1. For the Figures 1F and 1G, authors compared cellular compositions in Non-Malignant, AAH, AIS, MIA, and IA. Please provide statistical test results for the differences. Provide statistical tests for Figure S3A as well.
2. Are the differences in the cellular composition supported by individual patients? For example, AAH vs. MIA in Patient5; AAH vs. MIA vs. IA in Patient 6; etc. Statistical tests (major comment 1) and the support from individual patients will strengthen authors' conclusion on the cellular dynamics.
3. For the Figures 2C and 2G, please provide statistical tests for the differences in the cell composition. Also provide individual patient data as stated in the major comment 2.
4. For the Figures D and H, GSEA labels are not readable. Anyway, the authors stated that "...downregulated pathway was involved in inflammatory responses, such as the IFN- α and IFN- γ responses" in lines 160~162 ; "Cancer derived fibroblasts are associated...immune response, such as showing strong IFN- γ and IFN- α responses" in lines 179~181; and " Therefore, cellular dynamics in ECs and fibroblasts supported a consistent phenotypic shift.." in lines 181~183. Isn't it opposite phenotypic shift? Please explain or mend the sentences.
5. For the Figures S5B and S5C, please provide statistical tests for the differences in the cell composition. Also provide individual patient data as stated in the major comment 2.
6. In Figure 2K, authors performed trajectory analysis on mixed-lineage cells including B cells, NK cells, T cells, DCs, and Cancer cells, and then interpreted the results "CD8+ T and cancer cells were located in separate trajectory branches from the same ancestor, indicating their interactive differentiation states" in lines 199~201. It is an inappropriate analysis and the interpretation is wrong.
7. In line 218, what is the meaning of "CD141+DCs mediated by NK cells" ?
8. In lines 219~222, authors state that "cancer cells in myeloid cell clusters were mostly observed in late clinical stage IA tumors" and interpreted as "tumors communicating with immune cells after stage IA are capable of immune escape". Are they doublets? I don't find data supporting the authors' claim on this part.
9. What is the supporting evidence for the statement "Correlations among seven cell subtypes ...may reflect differences underlying the histopathology" in lines 230~232?
10. In the Characterization of epithelial cell lineages section, why did the authors use "AT2-like" nomenclature? Do all the AT2-like cluster cells show higher correlation to AT2 cells compared to the other epithelial cell types? Correlation matrix may help to delineate similarities between epithelial cell clusters.
11. The best part of this dataset is in the epithelial cell types, AT2 cells and their alterations in particular. I would recommend more comprehensive analysis on this part.
12. In Figures 3A and 3B, normal samples seem to contribute to the AT2 like-1, AT2 like-4, and AT2 like-5 clusters. Does it mean that those clusters are normal AT2 cells with an additional signaling activation, such as immune activation? Normal cells (both epithelial and stromal/immune) tend to form multi-patient clusters whereas cancer cells form patient-specific clusters without batch correction. Please check the patient contribution to each epithelial cluster.

13. The gene labels in Figure 3C are not readable and I cannot check the characteristics of each cluster.
14. In Figure 3D, CNA inference for combined epithelial cell types would be more appropriate to understand genetic aberrations in different cell type clusters. Matthew Meyerson group has demonstrated that arm-level CNA is less common in the pre/minimally invasive stages using bulk data (Nature commun. 2019 Nov29; 10(1):5472). Single cell level data in this study would further strengthen the observation. In the revised figure, provide color-codes for the cell types and patients/samples in addition to the stages.
15. How many samples were stained in Figure 3F, S7C, S8, and S9? Please provide statistics for the immunofluorescence data.
16. Please include AT2 cells from the non-malignant samples in the pseudotime analysis, Figure 4A. Also provide pseudotime analysis on the individual patients with multiple sampling. Pseudotime is reversed in Figure 4B.
17. For Figure 4D, do authors find similar alterations in the precancerous cells?
18. Regarding Figure 5B, is CellphoneDB run on combined dataset? Please provide results for each stage group, i.e. non-malignant, AAH, AIS, MIA, and IA. Are there any differences in the cellular interaction pairs or molecules among those groups?
19. For the Figure 5A, the number of significant interactions between cell types is very small. Are those associated with your modified p-value described in line 642 as "combined p-value of 0.001 (multiplying all p-values within each gene-pairs)"? Please explain more details.
20. Regarding Figure 5D (mistyped as 4D in line 401), it is difficult to agree with the authors' interpretation stated in lines 399~405. Autofluorescence and doublets could obscure flowcytometry data.
21. Regarding Figure 6A, are there significant differences between Normal vs. AIS/MIA for miRNA-10a and b-hydroxybutyric acid?
22. For the Figures 1C, S4B, S4D etc., the cells color-coded by "cancer" or "normal" showed distinct positions in the UMAP plot, indicating batch effects. If these data structure was intended by the author, please clarify why you are not applying batch correction. You may want to provide UMAP plots after batch corrections.

Minor comment

1. In Figure legend 3E, what is PDEC protocol?
2. Biomarker name for Figure 6A is missing.

Reviewer #2 (Remarks to the Author): Expert in lung cancer genomics and clinical research

The manuscript by Dr. Zhoufeng Wang et al. described the cellular landscape of TME of LAUD and its precursors by scRNA-seq, mIF, which is an important topic for lung cancer treatment and prevention. The authors have done a lot of work using different technologies, different specimens (human, animal; tissue, plasma etc.) and performed many interesting analyses. Unfortunately, there are several fundamental issues that make interpretation of the data difficult.

1. As the diagnosis of lung cancer precursors, particularly AAH and AIS requires careful pathologic assessment that requires FFPE processing. If the authors use half of the samples for single cell isolation, how can you be sure you are not missing the invasive component? For the samples subjected to scRNA seq, without pathology assessment, how do you know the histologic diagnosis was correct? How do you know the % of normal contamination? For example, AAH lesions are tiny, which makes them easily to have high proportion of normal lung tissues. Therefore, how do you know the difference you observed was not due to different among normal contamination in different stages? The authors should explain this and discuss this important limitation.
2. There is profound heterogeneity between different patients. With such a small sample size for each stage, the results are easily driven by one or two extreme cases. The authors should consider paired analyses of each patient to their matched normal and among different histologic stages, particularly in

patients with multifocal diseases.

3. By the same token of heterogeneity, what is the impact of other features like age, gender and smoking? Are the differences due to these different clinical features or due to different stage? The normal lung tissue data should be compared between smokers vs non-smokers, female vs male etc. This speaks again the importance of analysis of the tissues from the same patients.
4. What about genomic features? For example, oncodrivers are associated with distinct cancer biology. How can you be sure your observed difference was due to histologic stage or due to different driver status in different stages?
5. It will be more interesting to compare different stages and state the step-wise changes from normal lung to AAH, AIS, MIA and invasive cancers.
6. It is proposed that AAH may progress to AIS, MIA and then invasive lung adeno. However, this linear model may not be true for every lung nodule. The current study, and majority of previous studies are based on resected lung nodules, which is just one time point of cancer evolution. The AAH, AIS, MIA and invasive cancer in these studies are not the same group of cancer cells. They are actually independent primaries. The evolutionary trajectory from AAH, AIS, MIA etc. is inappropriate. This should be acknowledged and this limitation should be discussed.
7. The presentation and interpretation of the data is not clear in many sessions, which makes it difficult to follow the story. Examples are: A. Line 114: What do you mean "The proportion of T lymphocytes and myoid cells are highly reproducible across different patients."? The figures did not show individual patients and it is hard to see the reproducibility. B. Line 118: In tumor tissues, we discovered that AT2 feature cell types (AT2-like cells) were associated with malignant composition, and were present at the onset of LUAD development (Fig.1B and 1D). What do you mean by "composition"?

Following are some specific comments.

1. Line 158: The results revealed Myc targets as the top enriched signature in tumor ECs. Indeed, earlier studies indicated that c-Myc is essential for tumor angiogenesis, glycolysis and oxidative phosphorylation, all of which promote vessel sprouting. This is true, one important property of cancer cells. But these data is from EC cells, not cancer cells.
2. Except adding more effort and data, the part of plasma markers does not seem to be needed for the current manuscript.
3. There were no titles for the tables in the manuscript.
4. Line 109, how do you define AT2-like cell population? What are the differences and similarities with AT2 cells?
5. Line 114, please provide the UMAP visualization of all the cells identified by patients.
6. Line 127, for what reason did you included the data analysis from Lambrechts study? Why did you select 2 from the 8 patients? Are these the only adenocarcinomas? Why not other recent studies on lung adenocarcinomas?
7. Line 157, the GSEA should be performed separately for every patient that has matched samples of both tumor tissue and distant normal tissues. And common enriched signatures could be revealed accordingly. The reason is heterogeneity and small sample size as mentioned above.
8. Lines 189-191, it is hard to get your conclusion that CD8 and Treg were enriched but CD4 and NK were depleted during tumor progression only by combing AAH, AIS, MIA and IA as "cancer" then compare normal and these lesions. Please separate different stage lesions and provide p values. Also, if you want to investigate mechanism of immune evasion, only by looking at those cells in general is not enough, and more work could be done. Eg. CD4/CD8, CD8 CTL, Th CTL, Treg, Th2/Th1, Th17/Th1, M1, M2, etc.
9. Lines 291-222, what do you mean "late clinical stage IA tumors"?
10. Line 227, "An increase in the CD45+ population". Is the difference statistically different? Please provide p value.
11. Line 229-230, are there any correlation of gene expression with stage? eg. In more advanced stage LUAD, higher/lower expressions of some genes?
12. Lines 406-442: "Quantitative detection of key biomarkers in plasma of different stage LUAD", it is hard to make the conclusion according to the data you provided. Please also explain why you only

included normal, AIS and MIA when comparing CEA, miRNA-10a and beta-hydroxybutyric acid, however you included AIS, MIA, stage 1, 2 and 3 when comparing CEA, MDK and TIMP1. One can always identify "biomarkers". It does not mean much unless it can be validated on independent cohort. This part is not needed for the current story anyway in my opinion.

13. Line 433, please indicated the full name of TIMP1 and MDK the first time it appeared in the article.

14. Line 433, please double check if it is Fig.3C, rather Fig.3B, that reveals the gene expression patterns.

15. Line 435, it is Fig.6B, rather Fig.6A ...

16. Line 439, it is Fig.6C, rather Fig.6B ...

17. Table 1: Histologic stage should be included.

18. Figure 1: legend is not clear, for example the color map for B and C. please use different colors to distinguish NK cell, T cell and B cell clusters.

19. Figure 1C, please clarify "cancer/normal".

20. Figure 2D and H, please separate different stages.

21. Figure 3F, please quantify expression level of SFTPC.

22. Figure 6A, please label your figure.

23. Figure 6A, why no IA?

24. Fig 2K, N, it is arguable whether this is appropriate to construct the transcriptomic trajectory by using different cell types from different patients with no genetic relationship. What if you do it on individual patient level?

Reviewer #3 (Remarks to the Author): Expert in lung cancer, stem cells, and mouse models

In this paper, the authors present scRNA-Seq data from human LUAD samples, ranging from precancerous lesions such as AAH, to stage IV. The focus of the paper is on the early stages (AAH-stage IA). The authors did not enrich for any cell type, so that both epithelial cells and stromal cells are present in their analysis. They describe the main transcriptional findings of each cell type and analyze potential cell-cell interactions with bioinformatic tools. Finally, they identify early-stage LUAD markers that can be detected in patient plasma samples. Overall, the study of early-stage lung cancer is of increasing interest to the community. Most studies focus on late-stage cancer, and scRNA-Seq data-sets of the early-stages are few, which makes this data set of value to the community. However, many conclusions drawn in this publication are not supported by the data, and conclusions are overstated. Moreover, the paper lacks a clear focus. It covers a vast variety of cell types, includes one mouse model experiment, looks at cell-cell interactions, and secreted factors for early detection. The way it is presented, it could be a resource paper without in depth experimental validation.

Major points

- The wording of the text needs to be more precise. It is often confusing and not clear what the authors are trying to say. For instance, on page 4 lines 83-58 they refer to the stages AAH, AIS, MIA, and stage IA as 'major subtypes of LUAD'. It is not always clear what is referred to when the words subtype or stage are used throughout the text. Also, stage IA is often referred to as late-stage, which is confusing. Readers will associate late-stage LUAD with stages III and IV. Table 4 actually has two rows that read "IA(stage II)" and "IA (stage III and IV)". The list of patients sequenced also shows that stage III and IV patients were sequenced, but in the text it says the focus is on AAH-IA. However, the last paragraph actually talks about stage III. With so much conflicting information it is difficult to know what stage is represented in which figure.

- The citations need to be worked on. They are often not showing what was claimed in the text. For instance page 3: "Some of these mouse studies have been utilized in lung cancer lineage tracing; however, this could not be performed in human⁸." Reference number 8 does not include lineage tracing and is not relevant to this claim. Other references are missing, for instance when using genes

to annotate cell clusters, there should be a reference for that (e.g. line 154-156). Also, both Laughney et al and Dost et al contain scRNA-Seq of human LUAD and should be discussed in the introduction or discussion (references below).

- Some conclusions are drawn that are not fully supported by data. One example is line 227-237. The correlations in S6 are not very convincing, even though statistical significance is reached. But even if there is a correlation, correlation doesn't show causation. The authors should be more careful in drawing conclusions. If they speculate, it should be clear that it's speculation.

- Line 268: 'In IA tumors, SFTPC gene expression was almost undetectable'. This is a quite striking phenotype and could be explored more. Many labs have shown that LUAD expresses high levels of other AT2 markers such as Napsin-A, NKX2-1, SP-A1, or SP-H. Stainings of some of these markers should be included.

Minor points

- Figures 2D+H, 3D, and 5B are not legible

- Line 219-222, very confusing sentence. What are "cancer cells in myeloid cell clusters"? And how does this observation indicate that "immune cells after stage IA are capable of immune escape"?

- 3a shows AT2 or AT2-like clusters. This needs to be explained because it is inconsistent with other published scRNA-seq data (see Travaglini et al). What is difference between the clusters? Was only SFTPC expression used to annotate them? Annotation should not depend on only one gene.

- Information regarding the mutation status of the patients should be included. What are the driver mutations, EGFR, KRAS, others?

- Line 275-285: This experiment talks about mouse AT2 cells that get infected with an oncogenic EGFG variant in vivo. Decreasing SPC gene expression was observed. It was concluded: "Therefore, the results further supported that AT2 cells give rise to LUAD in spontaneous EGFRVIII mice and in human." The presented results do not support this conclusion. It is well established that AT2 cells can be a cell of origin of LUAD in mice and many papers can be cited for that. Furthermore, it is known that LUAD cells lose their AT2 lineage identity over time. Since the human studies presented in this paper focus on early stages, this experiment is unnecessary and does not add anything.

- Line 305 and 369: "negative regulation of the cell death response to stimulus (AQP5)28". Aquaporin 5 is commonly expressed in AT2 and AT1 cells. The paper cited here talks about it being a stem cell marker in the distal stomach which should not be relevant here.

- Line 379-381: Unclear wording. Needs to be explained better.

- Line 399-405: Not clear what this is showing. FACS shows that there is an increasing number of Epcam+ CD45+ cells in tumors as they progress. How does this lead to this conclusion: "thus confirming that immune cells communicating tumor cells towards immune suppression in LUAD"?

- Line 453: "Mechanistically, we have shown that alteration in the immune response during tumor progression is associated with immune-suppression via immune escape." No mechanistic studies that show this are presented.

Dost, A. F., Moye, A. L., Vedaie, M., Tran, L. M., Fung, E., Heinze, D., Villacorta-Martin, C., Huang, J., Hekman, R., Kwan, J. H., et al. (2020). Organoids Model Transcriptional Hallmarks of Oncogenic KRAS Activation in Lung Epithelial Progenitor Cells. *Cell Stem Cell*, 27(4):663–678.e8.

Laughney, A. M., Hu, J., Campbell, N. R., Bakhoun, S. F., Setty, M., Lavallée, V. P., Xie, Y., Masilionis, I., Carr, A. J., Kottapalli, S., et al. (2020). Regenerative lineages and immune-mediated pruning in lung cancer metastasis. *Nature Medicine*, 26(2):259–269.

Travaglini, K. J., Nabhan, A. N., Penland, L., Sinha, R., Gillich, A., Sit, R. V., Chang, S., Conley, S. D., Mori, Y., Seita, J., et al. (2020). A molecular cell atlas of the human lung from single-cell RNA sequencing. *Nature*, 587(7835):619–625.

REVIEWER COMMENTS

Reviewer #1 (Remarks to the Author): Expert in single-cell RNA-seq and lung cancer genomics

Review comment

In this study, Wang, Li, Zhou, and colleagues applied single-cell RNA sequencing to precancerous lesions of lung adenocarcinoma and characterized stepwise alterations from the non-malignant lung towards atypical adenomatous hyperplasia (AAH), adenocarcinoma in situ (AIS), minimally invasive adenocarcinoma (MIA), and invasive adenocarcinoma (IA). As the authors claimed, the study provides a valuable resource for the understanding of early events during the development of lung adenocarcinoma. However, most of the results are presented without statistical tests and sometimes flawed by interpretation of trajectory analysis on multi-lineage cells (cancer and immune cells) or potential doublets or auto-fluorescent cells. Over-interpretation and statements without basis afflict the entire manuscript.

Author's Responses: We appreciate the reviewer's recognition of the significance of our study. In the revision, we have provided additional results of new statistical tests as requested by reviewers. According to the reviewers' suggestions, we have revised some inaccurate and ambiguous statements and removed unnecessary content. Furthermore, to perform a deeper analysis on the epithelial cells, which distinguished this study from previous lung cancer single-cell studies, we have added an analysis on gene regulatory modules based on SCENIC to assess which transcription factors account for the differences in expression between epithelial subgroups and supplemented the mutation status information of the patients.

Major comments

1. For the Figures 1F and 1G, authors compared cellular compositions in Non-Malignant, AAH, AIS, MIA, and IA. Please provide statistical test results for the differences. Provide statistical tests for Figure S3A as well.

Author's Responses: We have included statistically significant results in Figure S5C and Figure S10 for Figures 1F, 1G and S3A. We gathered the cell fractions for individual patients, ran Kruskal-Wallis rank test (Laughney et al. Nature Medicine 2020. Ref 16) by comparing different tumor types against normal, as shown in Table 3. The original Fig 1F/G are removed to Figs. S5A/5B in the revised manuscript.

2. Are the differences in the cellular composition supported by individual patients? For example, AAH vs. MIA in Patient5; AAH vs. MIA vs. IA in Patient 6; etc. Statistical tests (major comment 1) and the support from individual patients will strengthen authors' conclusion on the cellular dynamics.

Author's Responses: We thank the reviewer for highlighting this point. We gathered the cell fractions for individual patients in Table 3. There is not enough number of cells from individual patient such as AAH and AIS. So direct comparison of cellular composition is not feasible with uneven cell numbers to obtain statistically significant conclusions.

3. For the Figures 2C and 2G, please provide statistical tests for the differences in the cell composition. Also provide individual patient data as stated in the major comment 2.

Author's Responses: We have included statistically significant results in Figure S5C for the two figures with the same statistical method above in the revised manuscript. We also added the individual-patient data in Table 3.

4. For the Figures 2D and 2H, GSEA labels are not readable. Anyway, the authors stated that "...downregulated pathway was involved in inflammatory responses, such as the IFN- α and IFN- γ responses" in lines 160~162; "Cancer derived fibroblasts are associated...immune response, such as showing strong IFN- γ and IFN- α responses" in lines 179~181; and "Therefore, cellular dynamics in ECs and fibroblasts supported a consistent phenotypic shift." in lines 181~183. Isn't it opposite phenotypic shift? Please explain or mend the sentences.

Author's Responses: We have improved the figure resolution, and moved Figures 2D and 2H to Figures 2C and 2F. We apologize for the confusion on the wording. We have revised these sentences in lines 160-165: "The top enriched signature in tumor ECs included Myc targets and the interferon (IFN) pathway. The c-Myc protein is essential for tumor angiogenesis, glycolysis and oxidative phosphorylation, all of which promote vessel sprouting. Upregulation of IFN- γ and IFN- α pathways is associated with inflammatory responses, so these processes may play a role in ECs biology".

5. For the Figures S5B and S5C, please provide statistical tests for the differences in the cell composition. Also provide individual patient data as stated in the major comment 2.

Author's Responses: We have included statistically significant results of two figures in Figure S5C and Figure S9B in the revised manuscript. We also gathered the cell fractions for each sample, ran Kruskal-Wallis rank test (Laughney et al. Nature Medicine 2020. Ref 16) by comparing different tumor types against normal, as shown in Table 3. But there is not enough number of cells from individual patient such as AAH and AIS to obtain statistically significant conclusions.

6. In Figure 2K, authors performed trajectory analysis on mixed-lineage cells including B cells, NK cells, T cells, DCs, and Cancer cells, and then interpreted the results “CD8⁺ T and cancer cells were located in separate trajectory branches from the same ancestor, indicating their interactive differentiation states” in lines 199~201. It is an inappropriate analysis and the interpretation is wrong.

Author's Responses: We agree with this comment. The only conclusion we could draw from trajectory analysis that lymphocytes were not associated with changes in the tumor biology. We have deleted Figure 2K and Figure 2N.

7. In line 218, what is the meaning of “CD141⁺DCs mediated by NK cells”?

Author's Responses: We have revised the sentence “CD141⁺ DCs mediated by NK cells” as “CD141⁺DCs, which express lymphotoxin beta transcripts in lung tumor tissues and contribute to tertiary lymphoid structure formation, were significantly less abundant in LUAD tissues than in normal lung” in lines 200-202 in the revised manuscript.

8. In lines 219~222, authors state that “cancer cells in myeloid cell clusters were mostly observed in late clinical stage IA tumors” and interpreted as “tumors communicating with immune cells after stage IA are capable of immune escape”. Are they doublets? I don't find data supporting the authors' claim on this part.

Author's Responses: We apologize for an error in this part. There were some mistakes in the sentence “The cancer cells in myeloid cell clusters were mostly observed in late clinical stage IA tumors”. We have removed the sentence in the revised manuscript. The conclusion of this part has been modified the sentences in lines 206~208 as follow:

“We speculate that the TMs in the IA stage induce tumor angiogenesis, promote tumor migration, invasion, and form an immunosuppressive TME”.

9. What is the supporting evidence for the statement “Correlations among seven cell subtypes ...may reflect differences underlying the histopathology” in lines 230~232?

Author’s Responses: We have removed the inaccurate sentence from the original manuscript, and modified the sentences as follow: “In fact, we found that the levels of seven cell subtypes, including CD4 T cells and T follicular helper cells, correlated with survival of LUAD patients” in lines 215~217.

10. In the Characterization of epithelial cell lineages section, why did the authors use “AT2-like” nomenclature? Do all the AT2-like cluster cells show higher correlation to AT2 cells compared to the other epithelial cell types? Correlation matrix may help to delineate similarities between epithelial cell clusters.

Author’s Responses: We thank the reviewer for suggestion. We have calculated the coefficient correlation between different cell clusters (Table 4) and epithelial types (Table 5). We called those cells AT2-like based on two lines of evidences: (1) similarity of gene expression; (2) CNV profiles. Clearly the AT2-like cells have the highest correlation with AT2 cells. In addition, based on the CNV analysis in Figure 3D, AT2 cells and the tumor epithelial cells show a gradient of chromosomal gain/loss events, not seen in other epithelial cell subtypes. This is separate piece of genetic evidence supporting the AT2 origin. For these reasons, we thought it was appropriate to label the tumor epithelial cells as “AT2-like” cells.

11. The best part of this dataset is in the epithelial cell types, AT2 cells and their alterations in particular. I would recommend more comprehensive analysis on this part.

Author’s Responses: We agree that the most valuable aspect of our data set is the epithelial component. In the revision, we have added new results on active transcription factor regulons in single cells of epithelial cell types. We presented SCENIC analysis in lines 247~260 “We employed single-cell regulatory network inference and clustering (SCENIC) to assess differences in expression of transcription factors (TFs) in epithelial cells. We found that RFX family motifs were highly activated in ciliated cells, and that MYC and HES1 were highly upregulated in basal cells (Fig. 3F). The TFs SPDEF, RBPJ, and CREB3 were activated in club clusters. Canonical AT1 TFs such as MYRF were

expressed in AT1 cells. Notably, we found that the lung fate TFs Nkx2-1 and the AT2 cell identity TFs Etv5 were enriched in the normal AT2 cell cluster. In contrast, the TFs HSF1, CEBPG and FOXQ1 were highly upregulated in AT2-like cells. HSF1 overexpression is common in lung cancer and correlates with tumor angiogenesis, while HSF1 activation in early-stage lung cancer cells and stroma are associated with poor outcome. The TF CEBPG correlates with antioxidant and DNA repair genes in airway epithelium and is associated with predisposition to lung cancer. Thus, our findings and the literature suggest that TFs of AT2 and AT2-like cells help drive LUAD tumorigenesis and progression.”.

12. In Figures 3A and 3B, normal samples seem to contribute to the AT2 like-1, AT2 like-4, and AT2 like-5 clusters. Does it mean that those clusters are normal AT2 cells with an additional signaling activation, such as immune activation? Normal cells (both epithelial and stromal/immune) tend to form multi-patient clusters whereas cancer cells form patient-specific clusters without batch correction. Please check the patient contribution to each epithelial cluster.

Author’s Responses: We thank the reviewer for this good suggestion. We have added Figure S14A that showed patient-specific contributions of cells to the five AT2 like clusters. As the reviewer pointed out above, normal samples indeed contribute to the AT2 like clusters. There would be two reasons: In Travaglini’s recent publication, they uncovered two clusters of AT2 cells of normal lung cell atlas. One cluster expressed higher levels of some canonical AT2 markers (SFTPA1, SFTPC and ETV5). The other, AT2-signalling cells, could be alveolar stem cells, similar to the rare, Wnt-active subpopulation of AT2 cells (Nature 2020 Ref.17). This result provided a consistent phenotype with our outcome that some AT2-signalling cells contribute to tumor cells. On the other hand, those “normal samples” are indeed “tumor adjacent normal tissues”, so they might contain low fractions of tumor cells.

13. The gene labels in Figure 3C are not readable and I cannot check the characteristics of each cluster.

Author’s Responses: We have revised the Figure 3C to provide clearer gene labels.

14. In Figure 3D, CNA inference for combined epithelial cell types would be more appropriate to understand genetic aberrations in different cell type clusters. Matthew

Meyerson group has demonstrated that arm-level CNA is less common in the pre/minimally invasive stages using bulk data (Nature comm. 2019 Nov29; 10(1):5472). Single cell level data in this study would further strengthen the observation. In the revised figure, provide color-codes for the cell types and patients/samples in addition to the stages.

Author's Responses: We have included the results of CNA analysis of all epithelial cell types combined in Figure S14B. Nonetheless, we feel that separating out the five epithelial cell types as we showed in Figure 3D and Figure S15 better illustrate a key message that the majority of tumor cells are present in the AT2/AT2-like groups, and hence it represents a genetic evidence supporting AT2 being the tumor origin. On the other hand, the scRNA-seq data are very sparse and only predict CNA at the chromosomal arm level. The findings related to 1Mb or smaller CNV by Matthew Meyerson group were made on bulk WES/WGS data, which have much denser genome coverage and higher chromosomal resolution. scRNA-seq data are not suitable for that type of analysis.

15. How many samples were stained in Figure 3F, S7C, S8, and S9? Please provide statistics for the immunofluorescence data.

Author's Responses: We performed immunofluorescence staining on normal and different histologic stage LUAD samples. The relative expression of different genes calculated in three samples. We have provided immunofluorescence statistics for the Figures 2K, 3G, 4E, 5D, S16, S18 and S20 in Fig.S12, Fig. S17 and Fig. S19.

16. Please include AT2 cells from the non-malignant samples in the pseudotime analysis, Figure 4A. Also provide pseudotime analysis on the individual patients with multiple sampling. Pseudotime is reversed in Figure 4B.

Author's Responses: In the revision, we have added the AT2 cells in the revised Figure 4A. And the reversed pseudotime has been corrected.

17. For Figure 4D, do authors find similar alterations in the precancerous cells?

Author's Responses: We thank the reviewer for highlighting this point. Even though we sequenced more samples and more single cells than other related studies (Table 6), we still did not have not enough precancerous cells in our data set for this analysis.

18. Regarding Figure 5B, is CellphoneDB run on combined dataset? Please provide results for each stage group, i.e. non-malignant, AAH, AIS, MIA, and IA. Are there any differences in the cellular interaction pairs or molecules among those groups?

Author's Responses: As the reviewer has pointed out, it is important to employ CellphoneDB for the accurate stage analysis. Therefore, we have presented the expression maps and dot plots of different stages to depict the interactions among the major cell types in Figure S21 and Figure S22. We have added the sentences as follow "The results were consistent with our comparison of expression patterns between normal tissue and tumors in different histologic stages (Figs. S21 and S22). The expression pattern of the FN1-A4B1 (A4B7) and ANXA-FPR1(FPR3) receptor-ligand complex indicates the existence of functional interactions between AT2-like cells and immune cells in IA stage." in lines 373~377.

19. For the Figure 5A, the number of significant interactions between cell types is very small. Are those associated with your modified p-value described in line 642 as "combined p-value of 0.001 (multiplying all p-values within each gene-pairs)"? Please explain more details.

Author's Responses: Yes. We want to be as stringent as possible to eliminate a large number of false positives showing up on the plot, and focus on the ones with significant differences. After calculating the p-value of the gene-pair interactions between different cell types, we multiplied the p-value across all the cell types and set the threshold to be at or below 0.001 eliminating the gene-pairs that are likely to be the noise or false-positives. As a result, only the gene-pair interactions between certain cell types that are statistically significant were included in the plot.

20. Regarding Figure 5D (mistyped as 4D in line 401), it is difficult to agree with the authors' interpretation stated in lines 399~405. Autofluorescence and doublets could obscure flowcytometry data.

Author's Responses: We thank the reviewer for suggestion. We focused on physical interactions between immune cells and epithelial cells in this part. We have removed the FACS results, and MOTiF™ PD-1/PD-L1 predesigned panel kit was used to simultaneously label regulatory T cells (FoxP3), epithelial cell positive (Pan CK), cytotoxic T cells (CD8), tumor-associated macrophage (CD68), and immune check

point markers (PD-1/ PD-L1) on the same tissue slide. We could only find that infiltration of immune cells increased with LUAD progression, and that PD-L1 was expressed more in tumor cells than in normal tissues. However, we did not detect clear interactions between tumors and immune cells because of intertumoral heterogeneity. We have revised this part as follow “To observe signaling crosstalk between immune cells and epithelial cells, normal and tumor tissues were stained for simultaneous detection of cytokeratin (tumor cells), CD8 (cytotoxic T cells), FoxP3 (regulatory T cells), CD68 (macrophages), PD-1, and PD-L1 (**Table 7, Figs. 5D and S23**). The results confirmed that infiltration of immune cells increased with LUAD progression, existing in a high spatiotemporal intertumoral heterogeneity in IA stage, and that PD-L1 was expressed more in tumor cells than in normal tissues. This is consistent with clinical observations that greater intertumoral heterogeneity is associated with lower treatment efficacy. However, we did not detect clear interactions between tumors and immune cells (**Fig. 5D**). Our results imply that greater efforts should be made to design cancer treatments that take into account not only the composition but also the heterogeneity and plasticity of the tumor ecosystem, as well as interactions within TME compartments and how they vary during cancer progression.” in lines 389~401.

21. Regarding Figure 6A, are there significant differences between Normal vs. AIS/MIA for miRNA-10a and b-hydroxybutyric acid?

Author’s Responses: Following the reviewer #2’s suggestion, we have removed this part from the revised manuscript.

22. For the Figures 1C, S4B, S4D etc., the cells color-coded by “cancer” or “normal” showed distinct positions in the UMAP plot, indicating batch effects. If these data structure was intended by the author, please clarify why you are not applying batch correction. You may want to provide UMAP plots after batch corrections.

Author’s Responses: We have already performed batch correction on the UMAP plot in the manuscript. Some of the cell clusters were highly enriched in cancer or normal due to the nature of the cell type, but not batch effect.

Minor comment

1. In Figure legend 3E, what is PDEC protocol?

Author's Responses: We apologize for an error in Figure 3E legend, we have now deleted “using PDEC protocol” in the legend.

2. Biomarker name for Figure 6A is missing.

Author's Responses: We have removed this part in the revised manuscript following the recommendation by the reviewer #2.

Reviewer #2 (Remarks to the Author): Expert in lung cancer genomics and clinical research

The manuscript by Dr. Zhoufeng Wang et al. described the cellular landscape of TME of LAUD and its precursors by scRNA-seq, mIF, which is an important topic for lung cancer treatment and prevention. The authors have done a lot of work using different technologies, different specimens (human, animal; tissue, plasma etc.) and performed many interesting analyses. Unfortunately, there are several fundamental issues that make interpretation of the data difficult.

1. As the diagnosis of lung cancer precursors, particularly AAH and AIS requires careful pathologic assessment that requires FFPE processing. If the authors use half of the samples for single cell isolation, how can you be sure you are not missing the invasive component? For the samples subjected to scRNA seq, without pathology assessment, how do you know the histologic diagnosis was correct? How do you know the % of normal contamination? For example, AAH lesions are tiny, which makes them easily to have high proportion of normal lung tissues. Therefore, how do you know the difference you observed was not due to different among normal contamination in different stages? The authors should explain this and discuss this important limitation.

Author's Responses: We agree with the reviewer's concern that there is a bias in the pathologic assessment and estimation of cellular compositions. As an alternative, we selected nodules that showed pure ground-glass characteristics by computed tomography, in order to reduce nodule heterogeneity in AAH, AIS, and MIA samples. Tumor specimens were cut along the largest diameter, and half were processed for intraoperative freezing and paraffin embedding, while the other half were carefully cut along the inner side of the nodule edge in order to avoid normal tissue. If we cut into normal tissue, the normal tissue was pruned away along with a small amount of the tumor to ensure that the single-cell sequencing sample was less contaminated by normal

cells. Tumor samples used for single-cell sequencing currently cannot be verified pathologically and cannot be assured of loss of invasive components, relying only on pathological evidence from the other half of the specimen. All samples were evaluated by two pathologists to determine pathologic diagnosis and tumor cellularity. In the revised manuscript, the CT scan and HE stained has been supplemented in Figure S2 and Figure S3.

We have added the sentences in Results section as follow: “We selected nodules that showed pure ground-glass characteristics by computed tomography, in order to reduce nodule heterogeneity in AAH, AIS, and MIA samples. Tumor specimens were cut along the largest diameter, and half were processed for intraoperative freezing and paraffin embedding, while the other half were carefully cut along the inner side of the nodule edge in order to avoid normal tissue. All samples were evaluated by two pathologists to determine pathologic diagnosis and tumor cellularity.” in lines 88~93.

We have added relevant sentences of this limitation in Discussion section as follow: “Taken together, our finding that LUAD progression involves loss of AT2 cell transcriptional identity and gain of stemness. The data also shed light on the underlying transcriptional signatures of distinct histologic stages of LUAD progression. However, the limitation of this study was that all samples were taken from primary tumors, which is only one point along the cancer trajectory, and because different tumors made up the sets of four histologic subtypes, so it was not a true longitudinal study. Our findings should be verified and extended in GEMM and PDX approaches, preferably with integration of genomics, transcriptomics and proteomics to capture lineage plasticity comprehensively. Importantly, how evolutionary selection acts during the acquisition of invasiveness warrants further research.” in lines 449~458.

2. There is profound heterogeneity between different patients. With such a small sample size for each stage, the results are easily driven by one or two extreme cases. The authors should consider paired analyses of each patient to their matched normal and among different histologic stages, particularly in patients with multifocal diseases.

Author’s Responses: We thank the reviewer for highlighting this point. With the limited number of cells present in few pair of patient samples, a direct analysis of individual paired is underpowered and not very meaningful. Nonetheless, we agree with the reviewer that this is an important point. To assess the possibility that we drew conclusions based on one or two extreme cases, we added the following analysis in this

revision. Focusing on the most interesting aspect of epithelial cells and tumor progression, as shown in the pseudotime plots in Figure 4A, we examined the contribution of individual patients. Specifically, we placed individual cells in one of the ten bins based on their “Component 1” values, which directly correlates the pseudotime, then we checked the contributions by individual patients, as well as their age, sex or other meta information to all the 10 bins. We have summarized the results of this analysis in the newly added Table 8.

3. By the same token of heterogeneity, what is the impact of other features like age, gender and smoking? Are the differences due to these different clinical features or due to different stage? The normal lung tissue data should be compared between smokers vs non-smokers, female vs male etc. This speaks again the importance of analysis of the tissues from the same patients.

Author’s Responses: This was a very valuable comment and we have integrated extensive analysis of patient information during the revision. Using the strategy outlined above to check the cell-number contribution by different features to each “pseudotime bin”, we observed representation of all clinical features along the pseudotime. It does not look obvious to us that one clinical feature is dominating or driving tumor progression. Nonetheless, we did notice more cells in the advanced stages for the non-smokers and those in the 40-49 age group (see below).

4. What about genomic features? For example, oncogenes are associated with distinct cancer biology. How can you be sure your observed difference was due to histologic stage or due to different driver status in different stages?

Author's Responses: We have performed WGS/WES on a subset of samples that have genomics DNA available. While driver mutations in EGFR and KRAS mutation detected in only six tumor samples (EGFR 746_750del, EGFR L858R, Kras G12C and G12D). We did not find evidence of one such features dominating (Figure S4 and Table 8). We have added a description of these results in lines 100-105: “We also performed whole exome sequencing (WES) or whole genome sequencing (WGS) on a subset of 26 tumor samples that have available DNA, and identified mutations in nine canonical driver genes, including BRAF, EGFR, ERBB2, HRAS, KRAS, MAP2K1, MET, NF1 and ROS1, while driver mutations in EGFR and KRAS mutation detected in only six tumor samples”.

5. It will be more interesting to compare different stages and state the step-wise changes from normal lung to AAH, AIS, MIA and invasive cancers.

Author's Responses: We thank the reviewer for highlighting this point. In the clinical setting, it is very difficult to collect large number of cells from very early stages such as AAH and AIS due to the nature of disease and clinical intervention guidelines. So

direct comparison of step-wise changes is not feasible with uneven cell numbers to obtain statistically significant conclusions. Therefore, we have determined that results in Figure 4A is more appropriate for the data set that we have.

6. It is proposed that AAH may progress to AIS, MIA and then invasive lung adeno. However, this linear model may not be true for every lung nodule. The current study, and majority of previous studies are based on resected lung nodules, which is just one time point of cancer evolution. The AAH, AIS, MIA and invasive cancer in these studies are not the same group of cancer cells. They are actually independent primaries. The evolutionary trajectory from AAH, AIS, MIA etc. is inappropriate. This should be acknowledged and this limitation should be discussed.

Author's Responses: We acknowledge that due to limited samples size, our findings are exciting but exploratory, and are therefore restricted to hypothesis generation. However, given the difficulty of obtaining such specimens, our current study represents a unique model of human lung cancer progression. We have revised the text according to the suggestion. We have revised the part in the discussion as follow: "Taken together, our finding that LUAD progression involves loss of AT2 cell transcriptional identity and gain of stemness. The data also shed light on the underlying transcriptional signatures of distinct histologic stages of LUAD progression. However, the limitation of this study was that all samples were taken from primary tumors, which is only one point along the cancer trajectory, and because different tumors made up the sets of four histologic subtypes, so it was not a true longitudinal study. Our findings should be verified and extended in GEMM and PDX approaches, preferably with integration of genomics, transcriptomics and proteomics to capture lineage plasticity comprehensively. Importantly, how evolutionary selection acts during the acquisition of invasiveness warrants further research." in lines 449~458.

7. The presentation and interpretation of the data is not clear in many sessions, which makes it difficult to follow the story. Examples are: A. Line 114: What do you mean "The proportion of T lymphocytes and myoid cells are highly reproducible across different patients."? The figures did not show individual patients and it is hard to see the reproducibility. B. Line 118: In tumor tissues, we discovered that AT2 feature cell types (AT2-like cells) were associated with malignant composition, and were present

at the onset of LUAD development (Fig.1B and 1D). What do you mean by “composition”?

Author’s Responses: We have removed this confusing sentence “The proportion of T lymphocytes and myeloid cells are highly reproducible across different patients”, while added the individual patient data in Table 3. We have changed the “we discovered that AT2 feature cell types (AT2-like cells) were associated with malignant composition, and were present at the onset of LUAD development as “Analysis of normal and tumor epithelial cell types revealed a group of cells (AT2-like cells) closely resembling AT2 cells that emerged initially within AAH.” in lines 111~112.

Following are some specific comments.

1. Line 158: The results revealed Myc targets as the top enriched signature in tumor ECs. Indeed, earlier studies indicated that c-Myc is essential for tumor angiogenesis, glycolysis and oxidative phosphorylation, all of which promote vessel sprouting. This is true, one important property of cancer cells. But these data is from EC cells, not cancer cells.

Author’s Responses: We partially agree with this comment. This observation was made by comparing tumor ECs with normal ECs. c-Myc is highly expressed in endothelial cells, and could promote cell growth and transformation, as well as vasculogenesis, by functioning as a master regulator of angiogenic factors. (Ref 18. Genes & Development 2002). Therefore, c-Myc is essential for vasculogenesis and angiogenesis during tumor progression.

2. Except adding more effort and data, the part of plasma markers does not seem to be needed for the current manuscript.

Author’s Responses: We have removed this part in the revised manuscript.

3. There were no titles for the tables in the manuscript.

Author’s Response: We have gone through the manuscript and added titles for all tables.

4. Line 109, how do you define AT2-like cell population? What are the differences and similarities with AT2 cells?

Author's Response: In the revision, the expression patterns of multiple marker genes were shown in Figure 1E and Figure 3C. We have added a separate dot plot for the epithelial genes (Figure S5) reported in the Krasnow et al. Nature 2020 paper (ref 17), and the results revealed that AT2-like gene features were close to AT2 cells. On the other hand, we calculated coefficient correlation using gene signatures of cell clusters. The cell-cell correlation matrix based on differentially expressed genes revealed that AT2-like cells are highly correlated to AT2 cells (Table 4 and Table 5).

5. Line 114, please provide the UMAP visualization of all the cells identified by patients.

Author's Response: We have provided the patient-specific cell numbers for each cluster in Table 3.

6. Line 127, for what reason did you included the data analysis from Lambrechts study? Why did you select 2 from the 8 patients? Are these the only adenocarcinomas? Why not other recent studies on lung adenocarcinomas?

Author's Response: As the reviewer has pointed out above, the main novelty of our manuscript is that we focused our studies on early histological stage of LUAD. We included data from other published study as a validation purpose, to demonstrate that the identification of the major cell types from our samples were reproducible in independent data. As a matter of fact, only 2 of 8 samples from the Lambrechts study (ref 14) were from patients with adenocarcinomas. We included them to make sure the sample type is consistent. By the time we were preparing our manuscript, there was only one other publication highly relevant to ours: <https://doi.org/10.1038/s41467-020-16164-1>. We manually checked the marker genes from this publication and found that they were consistent with ours and Lambrechts study. Furthermore, most of recent articles on lung adenocarcinoma single-cell sequencing were based on invasive adenocarcinoma. In contrast our samples included samples from earlier stages, such as AAH, AIS and MIA. In this study we wanted to focus on early histological stage of LUAD, so we did not include other recent studies into our data analysis. In order to compare this study with the recent published single cell sequencing studies on LUAD, we provided a summary in the revised table 6.

7. Line 157, the GSEA should be performed separately for every patient that has matched samples of both tumor tissue and distant normal tissues. And common enriched signatures could be revealed accordingly. The reason is heterogeneity and small sample size as mentioned above.

Author's Response: As suggested, we have repeated GSEA analysis on the paired samples that have enough cells (>300 cells per sample). The result was found that Myc targets and IFN pathway as the top enriched signature in tumor ECs for the most patients. We have revised the sentence in the revised manuscript "To learn more about the biology underlying these cell states, we used Gene Set Enrichment Analysis (GSEA) to compare expression profiles between tumor and normal ECs (**Figs. 2C and S11**). The top enriched signature in tumor ECs included Myc targets and the interferon (IFN) pathway. The c-Myc protein is essential for tumor angiogenesis, glycolysis and oxidative phosphorylation, all of which promote vessel sprouting. Upregulation of IFN- γ and IFN- α pathways is associated with inflammatory responses, so these processes may play a role in ECs biology. The endothelium represents the primary interface between circulating immune cells and the tumor, this may help explain how ECs contribute to LUAD" in lines 158~166.

8. Lines 189-191, it is hard to get your conclusion that CD8 and Treg were enriched but CD4 and NK were depleted during tumor progression only by combing AAH, AIS, MIA and IA as "cancer" then compare normal and these lesions. Please separate different stage lesions and provide p values. Also, if you want to investigate mechanism of immune evasion, only by looking at those cells in general is not enough, and more work could be done. Eg. CD4/CD8, CD8 CTL, Th CTL, Treg, Th2/Th1, Th17/Th1, M1, M2, etc.

Author's Response: Thanks for your suggestion, we have included statistically significant results in Figure S5C in different stage samples. However, as immune cells were not the focus of this study, we did not present more analysis on them. There are multiple published studies already, there is not much novel insights we can add. Covering too many different topics will make this paper unfocused. We have deleted the related parts in the revised manuscript.

9. Lines 291-222, what do you mean "late clinical stage IA tumors"?

Author's Response: We have modified the ambiguous words. The “stage IA tumors” means “invasive adenocarcinoma”. In the revision, we have used the “histological subtypes” to describe the AAH, AIS, MIA, and IA to avoid confusion with TNM stage.

10. Line 227, “An increase in the CD45⁺ population”. Is the difference statistically different? Please provide p value.

Author's Response: We performed immunofluorescence staining on normal and different histologic stage LUAD samples. The relative fluorescence intensity of different genes was calculated, and statistically significant results were shown in Fig. S12A. We have revised the sentence in the revised manuscript: “The CD45⁺ and fibronectin⁺ populations increased as the tumor progressed (**Fig. S12**), consistent with the idea that immune cell infiltration and EMT help define the TEM during tumor progression” in lines 213~217.

11. Line 229-230, are there any correlation of gene expression with stage? eg. In more advanced stage LUAD, higher/lower expressions of some genes?

Author's Response: This is a good point, and indeed we have tried to find gene expression signature that we are confident with. Such as JUN, TIMP1 and MDK were highly expressed at the IA stage, whereas LAMP3 were substantially diminished as LUAD progression in Figure 4C. To confirm the expression of identified genes, we performed immunofluorescent staining of TIMP1 and MDK of different stage LUAD in Figure 18A.

12. Lines 406-442: “Quantitative detection of key biomarkers in plasma of different stage LUAD”, it is hard to make the conclusion according to the data you provided. Please also explain why you only included normal, AIS and MIA when comparing CEA, miRNA-10a and beta-hydroxybutyric acid, however you included AIS, MIA, stage 1, 2 and 3 when comparing CEA, MDK and TIMP1. One can always identify “biomarkers”. It does not mean much unless it can be validated on independent cohort. This part is not needed for the current story anyway in my opinion.

Author's Response: Thanks for your suggestion, we have removed this part in the revised manuscript.

13. Line 433, please indicated the full name of TIMP1 and MDK the first time it appeared in the article.

Author's Response: We made the corrections in lines 299~303 as follow “Lastly, the genes in group 4 were involved in extracellular matrix organization (tissue inhibitor matrix metalloproteinase 1 TIMP1) as well as cell-cell signaling and regulation of cell migration (S100A4, VEGFR). We also identified several genes previously linked to cancer progression, such as midkine (MDK), SOX4 and LYZ”.

14. Line 433, please double check if it is Fig.3C, rather Fig.3B, that reveals the gene expression patterns.

Author's Response: We have corrected the error in the revised manuscript as Figure 4C.

15. Line 435, it is Fig.6B, rather Fig.6A ...

Author's Response: Thanks for your suggestion, we have removed this part in the revised manuscript.

16. Line 439, it is Fig.6C, rather Fig.6B ...

Author's Response: Thanks for your suggestion, we have removed this part in the revised manuscript.

17. Table 1: Histologic stage should be included.

Author's Response: Table 1 in our original submission already contained the histologic stage information. We have adjusted the format of table 1 for easy reading.

18. Figure 1: legend is not clear, for example the color map for B and C. please use different colors to distinguish NK cell, T cell and B cell clusters.

Author's Response: As suggested, we have updated the Figure B with different color to distinguish the NK cell, T cell and B cell clusters in the revised manuscript.

19. Figure 1C, please clarify “cancer/normal”.

Author's Response: The “normal” means the normal lung tissues from an adjacent region of tumor in the same lobe. We have modified “cancer” as “tumor” Figure 1C.

20. Figure 2D and H, please separate different stages.

Author's Response: Endothelial and fibroblast cells were enriched in tumor adjacent normal tissues and only small percentage of cancer cells. As a result, performing GSEA analysis on different stages would not be feasible.

21. Figure 3F, please quantify expression level of SFTPC.

Author's Response: In the revision, we have added quantification of the relative fluorescence intensity of SFTPC gene in the revised Figure S12B.

22. Figure 6A, please label your figure.

Author's Response: Thanks for your suggestion, we have removed this part in the revised manuscript.

23. Figure 6A, why no IA?

Author's Response: Thanks for your suggestion, we have removed this part in the revised manuscript.

24. Fig 2K, N, it is arguable whether this is appropriate to construct the transcriptomic trajectory by using different cell types from different patients with no genetic relationship. What if you do it on individual patient level?

Author's Response: We agree with this comment. The only conclusion we could draw from trajectory analysis that lymphocytes were not associated with changes in the tumor biology. We have removed Figure 2K and Figure 2N from the revision.

Reviewer #3 (Remarks to the Author): Expert in lung cancer, stem cells, and mouse models

In this paper, the authors present scRNA-Seq data from human LUAD samples, ranging from precancerous lesions such as AAH, to stage IV. The focus of the paper is on the early stages (AAH-stage IA). The authors did not enrich for any cell type, so that both epithelial cells and stromal cells are present in their analysis. They describe the main transcriptional findings of each cell type and analyze potential cell-cell interactions with bioinformatic tools. Finally, they identify early-stage LUAD markers that can be detected in patient plasma samples. Overall, the study of early-stage lung cancer is of

increasing interest to the community. Most studies focus on late-stage cancer, and scRNA-Seq data-sets of the early-stages are few, which makes this data set of value to the community. However, many conclusions drawn in this publication are not supported by the data, and conclusions are overstated. Moreover, the paper lacks a clear focus. It covers a vast variety of cell types, includes one mouse model experiment, looks at cell-cell interactions, and secreted factors for early detection. The way it is presented, it could be a resource paper without in depth experimental validation.

Major points

1. The wording of the text needs to be more precise. It is often confusing and not clear what the authors are trying to say. For instance, on page 4 lines 83-58 they refer to the stages AAH, AIS, MIA, and stage IA as ‘major subtypes of LUAD’. It is not always clear what is referred to when the words subtype or stage are used throughout the text. Also, stage IA is often referred to as late-stage, which is confusing. Readers will associate late-stage LUAD with stages III and IV. Table 4 actually has two rows that read “IA (stage II)” and “IA (stage III and IV)”. The list of patients sequenced also shows that stage III and IV patients were sequenced, but in the text it says the focus is on AAH-IA. However, the last paragraph actually talks about stage III. With so much conflicting information it is difficult to know what stage is represented in which figure.

Author’s Response: We agree with this comment. We have modified the confusing description about subtypes of LUAD. We used the ‘histological subtypes’ to describe the AAH, AIS, MIA and invasive adenocarcinomas (IA), which were classified by their histologic growth pattern. In the 2015 World Health Organization classification, AAH and AIS are both defined as preinvasive lesions, whereas MIA is identified as an early invasive adenocarcinoma that is not expected to recur if removed completely. The “clinical stage” means TNM stage (such as I, II, III and IV stage), which based on tumor size, lymph node, and metastasis. We have corrected all stages as “histological stage” in the revised manuscript.

2. The citations need to be worked on. They are often not showing what was claimed in the text. For instance page 3: “Some of these mouse studies have been utilized in lung cancer lineage tracing; however, this could not be performed in human ⁸.” Reference number 8 does not include lineage tracing and is not relevant to this claim. Other references are missing, for instance when using genes to annotate cell clusters, there should be a reference for that (eg. line 154-156). Also, both Laughney et al and

Dost et al contain scRNA-Seq of human LUAD and should be discussed in the introduction or discussion (references below).

Author's Response: Thanks for your suggestion. In the revision, we have added the two references in the introduction as “Ref 16 and Ref 17” in lines 70~72 and 121-122. We also changed Ref 8 (Marjanovic, et al. Cancer Cell. 2020) to Ref 7~9 (Cheung, et al. Oncogene 2015, Xu, et al. PNAS 2012, Mainardi et al. PNAS 2014) in lines 70~71. These three studies attempted to determine which of cells are targets for transformation in lung cancers by using genetically engineered mouse models (GEMMs).

3. Some conclusions are drawn that are not fully supported by data. One example is line 227-237. The correlations in S6 are not very convincing, even though statistical significance is reached. But even if there is a correlation, correlation doesn't show causation. The authors should be more careful in drawing conclusions. If they speculate, it should be clear that it's speculation.

Author's Response: We are thankful for this suggestion. As suggested, we want to identify key cell types would be correlation with LUAD progression in this part, so we have changed some speculative statements in the revised manuscript as follow: “In fact, we found that the levels of seven cell subtypes, including CD4 T cells and T follicular helper cells, correlated with survival of LUAD patients.” in lines 215~217.

4. Line 268: ‘In IA tumors, SFTPC gene expression was almost undetectable’. This is a quite striking phenotype and could be explored more. Many labs have shown that LUAD expresses high levels of other AT2 markers such as Napkin-A, NKX2-1, SP-A1, or SP-H. Stainings of some of these markers should be included.

Author's Response: Thanks for your suggestion. We have added other AT2 markers as NKX2-1 and SFTPA for protein fluorescent immunostaining in Figure S16B. Consistent with the reduced SFTPC expression, Nkx2-1 and SFTPA staining were specifically reduced from AAH to IA stage tumors. Two recent studies analyzed data from genetically engineered mouse models (ref 12. Cancer Cell 2020, ref 13. Cell Stem Cell 2020), revealed that a reduction in AT2 cell lineage marker gene expression is an early consequence of oncogenic KRAS. Our result also suggested that human LUAD evolution is characterized by a loss of the AT2 feature of the lung lineage and the emergence of an alternative dedifferentiated stem-like state.

Minor points

1. Figures 2D+H, 3D, and 5B are not legible.

Author's Response: We have revised the figures and edited the label size in Figure 2C, 2F, 3D, and 5B.

2. Line 219-222, very confusing sentence. What are “cancer cells in myeloid cell clusters”? And how does this observation indicate that “immune cells after stage IA are capable of immune escape”?

Author's Response: Thanks for your suggestion, we have removed these two confusing sentences as: “cancer cells in myeloid cell clusters” and “immune cells after stage IA are capable of immune escape” We have modified this part as follow: “Alveolar macrophages (AMs), which highly express MARCO, FABP4, and MCEMP1, were detected mainly in normal and early-stage LUAD; tumor macrophages (TMs) comprised the remaining tumor-enriched clusters and were present mostly in IA tumors (Figs. 2I and S9A). TMs showed high expression of SPP1, APOE, CCL2 genes, involved in apolipoprotein metabolism.” in lines 202~206.

3. 3a shows AT2 or AT2-like clusters. This needs to be explained because it is inconsistent with other published scRNA-seq data (see Travaglini et al). What is difference between the clusters? Was only SFTPC expression used to annotate them? Annotation should not depend on only one gene.

Author's Response: In Travaglini's recent publication (ref 17), they uncovered two clusters of AT2 cells of normal lung cell atlas. One cluster expressed higher levels of some canonical AT2 markers (SFTPA1, SFTPC and ETV5). The other, AT2-signalling cells, could be alveolar stem cells, similar to the rare, Wnt-active subpopulation of AT2 cells. This result provided a consistent phenotype with our outcome, which the expression patterns of canonical AT2 marker genes as SFTPA1, SFTPC and PGC in Figure 1E and Figure 3C. In our study, we focused our studies on the consequences of LUAD from AT2 cells, and calculated coefficient correlation using gene signatures of cell cluster. The cell-cell correlation matrix based on differentially expressed genes revealed that AT2-like cells are highly correlated to AT2 cells (Table 4 and Table 5). We also have added a separate dot plot for the epithelial genes (Figure S6) reported in the Travaglini et al. Nature 2020 paper (ref 17), and the results revealed that AT2-like gene features were close to AT2 cells.

Figure S5. Dot plot of differentially expressed epithelial features labeled based on cell clusters.

4. Information regarding the mutation status of the patients should be included. What are the driver mutations, EGFR, KRAS, others?

Author’s Response: We performed whole exome sequencing (WES) or whole genome sequencing (WGS) for 26 tumor samples that have DNA available, in order to identify nine commonly mutated lung cancer genes, including BRAF, EGFR, ERBB2, HRAS, KRAS, MAP2K1, MET, NF1 and ROS1. We have reported the EGFR and KRAS driver mutations in Figure. S4. We have added the sentences about genomic features as follow: “We also performed whole exome sequencing (WES) or whole genome sequencing (WGS) on a subset of 26 tumor samples that have available DNA, and identified mutations in nine canonical driver genes, including BRAF, EGFR, ERBB2, HRAS, KRAS, MAP2K1, MET, NF1 and ROS1, while driver mutations in EGFR and KRAS mutation detected in only six tumor samples” in lines 100-105.

5. Line 275-285: This experiment talks about mouse AT2 cells that get infected with an oncogenic EGFG variant in vivo. Decreasing SPC gene expression was observed. It was concluded: “Therefore, the results further supported that AT2 cells give rise to

LUAD in spontaneous EGFRVIII mice and in human.” The presented results do not support this conclusion. It is well established that AT2 cells can be a cell of origin of LUAD in mice and many papers can be cited for that. Furthermore, it is known that LUAD cells lose their AT2 lineage identity over time. Since the human studies presented in this paper focus on early stages, this experiment is unnecessary and does not add anything.

Author’s Response: We agree with this comment. We have removed this part in the revision.

6. Line 305 and 369: “negative regulation of the cell death response to stimulus (AQP5) 28”. Aquaporin 5 is commonly expressed in AT2 and AT1 cells. The paper cited here talks about it being a stem cell marker in the distal stomach which should not be relevant here.

Author’s Response: We have removed this inaccurate sentence.

7. Line 379-381: Unclear wording. Needs to be explained better.

Author’s Response: We have modified the sentence as follow: “The epithelial cell clusters, especially the AT2/AT2-like clusters, as well as the fibroblast clusters showed the most interactions with other cell types, such with myeloid cells (**Fig. 5A**). This suggests interaction between epithelial and stromal cells involving certain receptor-ligand gene pairs (**Fig. 5B**). We focused on gene pairs associated with $p < 0.001$ in strongly interacting cell types. AT2-like cells express high levels of ANXA1, MDK, and FN1 (**Fig. 5C**), the receptor of which (FPR1, SORL and a4b1) were expressed by DCs and macrophages. The results were consistent with our comparison of expression patterns between normal tissue and tumors in different histologic stages (**Figs. S21 and S22**). The expression pattern of the FN1-A4B1 (A4B7) and ANXA-FPR1(FPR3) receptor-ligand complex indicates the existence of functional interactions between AT2-like cells and immune cells in IA stage.” in lines 366~377.

8. Line 399-405: Not clear what this is showing. FACS shows that there is an increasing number of Epcam⁺ CD45⁺ cells in tumors as they progress. How does this lead to this conclusion: “thus confirming that immune cells communicating tumor cells towards immune suppression in LUAD”?

Author's Response: We thank the reviewer for suggestion. We focused on physical interactions between immune cells and epithelial cells in this part. We have removed the FACS results, and MOTiF™ PD-1/PD-L1 predesigned panel kit was used to simultaneously label regulatory T cells (FoxP3), epithelial cell positive (Pan CK), cytotoxic T cells (CD8), tumor-associated macrophage (CD68), and immune check point markers (PD-1/ PD-L1) on the same tissue slide. We could only find that infiltration of immune cells increased with LUAD progression, and that PD-L1 was expressed more in tumor cells than in normal tissues. However, However, we did not detect clear interactions between tumors and immune cells because of intertumoral heterogeneity. We have revised this part as follow “To observe signaling crosstalk between immune cells and epithelial cells, normal and tumor tissues were stained for simultaneous detection of cytokeratin (tumor cells), CD8 (cytotoxic T cells), FoxP3 (regulatory T cells), CD68 (macrophages), PD-1, and PD-L1 (**Table 7, Figs. 5D and S23**). The results confirmed that infiltration of immune cells increased with LUAD progression, existing in a high spatiotemporal intertumoral heterogeneity in IA stage, and that PD-L1 was expressed more in tumor cells than in normal tissues. This is consistent with clinical observations that greater intertumoral heterogeneity is associated with lower treatment efficacy. However, we did not detect clear interactions between tumors and immune cells (**Fig. 5D**). Our results imply that greater efforts should be made to design cancer treatments that take into account not only the composition but also the heterogeneity and plasticity of the tumor ecosystem, as well as interactions within TME compartments and how they vary during cancer progression.” in lines 389~401.

9. Line 453: “Mechanistically, we have shown that alteration in the immune response during tumor progression is associated with immune-suppression via immune escape.” No mechanistic studies that show this are presented.

Author's Response: This sentence was removed in the revision.

Additional References

Ref 13. Dost, A. F., Moye, A. L., Vedaie, M., Tran, L. M., Fung, E., Heinze, D., Villacorta-Martin, C., Huang, J., Hekman, R., Kwan, J. H., et al. (2020). Organoids Model Transcriptional Hallmarks of Oncogenic KRAS Activation in Lung Epithelial Progenitor Cells. *Cell Stem Cell*, 27(4):663–678.e8.

Ref 16. Laughney, A. M., Hu, J., Campbell, N. R., Bakhoun, S. F., Setty, M., Lavallée,

V. P., Xie, Y., Masilionis, I., Carr, A. J., Kottapalli, S., et al. (2020). Regenerative lineages and immune-mediated pruning in lung cancer metastasis. *Nature Medicine*, 26(2):259–269.

Ref 17. Travaglini, K. J., Nabhan, A. N., Penland, L., Sinha, R., Gillich, A., Sit, R. V., Chang, S., Conley, S. D., Mori, Y., Seita, J., et al. (2020). A molecular cell atlas of the human lung from single-cell RNA sequencing. *Nature*, 587(7835):619–625.

REVIEWER COMMENTS

Reviewer #1 (Remarks to the Author):

Reviewer 1 (2nd round review)

The manuscript is slightly improved compared to the original submission in terms of research focus and statistical tests have been provided where applicable. Still, there are poor usage of figures and tables, and disparities between figures and texts. Figure rearrangement is highly recommended to convincingly demonstrate the cellular alterations during tumor progression from normal>>AAH>>AIS>>MIA>>stage IA LUAD.

Below are specific comments to the revised figures and texts.

1. In figure 1B~D: presentation of Cluster info and then Normal through IA tumor in the same format and preferably side by side will clearly show the change of cell type abundance in the UMAP space.(Suggestion)
2. Figure 1G in the original submission (revised S5B) along with stats (revised S5C) clearly demonstrate the alterations in immune cell dynamics. Current Figure 1E can be supplementary. (Suggestion)
3. Individual patient data in the revised figure S5A is incomprehensible. Different format or reordering by analysis group (normal, AAH, AIS, MIA, IA tumor) will increase the visibility.
4. Revised line 116-117, where is the matching data?
5. Revised Table 5 is redundant with Table 4. Use of the epithelial cell type marker genes (DEGs for Basal, Ciliated, Club, AT2 like, AT2, AT1) will be better to demonstrate AT2 likeness. Also figure format presentation of Table 5 seems to be more suitable along with revised Figure S6.
6. Revised figure S6: demonstration of more markers (for Basal, Ciliated, Club, AT2 like, AT2, AT1) will be more convincing.
7. Results section 2 and revised figure2 focus on the normal vs. tumor analysis, not on the transitions at normal>>AAH>>AIS>>MIA>>stage IA LUAD. The normal-tumor analysis is mostly reiteration of previous publications.
8. Lines216-218: Description of the Figure S13 needs to be more specific. Gene expression levels for CD4+ T, T follicular helper, Activated DC, and Granulocyte have a negative association where as those for plasmacytoid DC, CD141+ DC, and DC have a positive association with the survival period.
9. Figure 3D is redundant with Supplementary Figure S14B.
10. Figure 3E: Grouped (AAH>>AIS>>MIA>>stage IA LUAD) figure representation would be easier to comprehend.
11. Figure 3G: Does IA LUAD include tumors with a lepidic growth pattern?
12. Figure 4A: The opposite direction of AAH and AIS vs. MIA and IA from the normal AT2 cells in the trajectory contradicts the authors' claim of gradual changes from normal to AAH>>AIS>>MIA>>stage IA LUAD. Why pseudotime 0 is assigned to the IA stage tumor cells?
13. Line 392-395: Table 7 is the list of antibodies used and statistical tests were performed between pan-CK+ vs. pan-CK- cells, not between the tissue groups (Normal, AAH, AIS, MIA, stage IA LUAD) in the Supplementary Figure S23. How come these results confirmed that infiltration of immune cells increased with LUAD progression?

Reviewer #2 (Remarks to the Author):

I appreciate the efforts that the authors made to address the reviewers' questions and comments. I think the data generated are valuable for the field and most of the analyses are reasonable. The authors should be congratulated for doing such an important and challenging study.

The comments from me and the other reviewers are mainly that many findings are overstated. Study on early carcinogenesis is difficult. There are a lot of limitations of the current study and many other previous studies, as exemplified below.

1. Profound heterogeneity of the lung cancers and precancers in different patients. This may be the main reason why many studies get different and even contra-indicatory results. Some recent studies such as PMID: 33083004; PMID: 33571124; PMID: 33976164 for example, should be discussed in the context of the findings from the current study.
2. The challenges of getting single cells from these early-stage diseases. As the authors mentioned, parried analyses could not be done due to small number of cells from single samples. For single cell seq studies on cancers, people argue that samples with less than 3000 cells are not informative and should be removed. Some of the samples have way less cells than that. If the sample size is very big, this may not be a major issue since these could be balanced out. But given the small sample size, some findings could be driven by outliers.
3. The controversial diagnosis of these entities: I don't think using two pathologists and focusing on GGOs will solve the problem of diagnosis of AAH, AIS and MIA. These have to be diagnosed by examining the whole section and many pathologist experts advocate for FFPE only and WHOLE section. There is no good solution at this time for such studies and the authors did the best they can. But you should discuss about this technical issue as the limitation for the field rather say you have solved the problem by focusing on GGO and using two pathologists.

The authors should clearly point out these limitations so future readers don't take these data interpretation as the "truth". These challenges will also be valuable for the whole field to brain storming.

Reviewer #3 (Remarks to the Author):

The authors have responded to each of the reviewers' comments and have made changes in the manuscript accordingly. Most notably, the mouse data, plasma data, and FACS data were removed, while important patient data, including analysis of driver mutations, and immunofluorescence staining was added. The manuscript now has a clearer focus on the progression of human early-stage LUAD epithelial cells and the contributions of the tumor microenvironment. I agree with the authors' comment that the findings "are exciting but exploratory, and are therefore restricted to hypothesis generation." However, due to the paucity of published human early-stage LUAD data and the difficulties associated with acquiring early-stage patient samples I think it will be important to publish this study and make it available to the research community after some minor revisions.

Comments:

1. With the normal cells present in the pseudotime analysis in figure 4A, the data does not convincingly show a linear progression from normal to AAH, AIS, MIA, and stage IA. Also, the authors did not update the main text after including the normal cells. In the heatmap in 4B, it looks like MIA cells, not stage IA cells, make up the right part of the heatmap. As another reviewer brought up, this linear relationship cannot be assumed and I wonder if this type of trajectory analysis is appropriate for this dataset, or if it should be left out. At the very least, the authors should comment on the limitations of using Monocle trajectory analysis on this dataset and explain why the normal cells line up the way they do in figure 4A.
2. "GSEA comparing fibroblasts from normal and tumor tissues showed that cancer-derived fibroblasts were associated with the epithelial-mesenchymal transition (EMT) and with strong IFN- γ and IFN- α responses¹⁵ (Fig. 2F)" (line 179-181) - Actually, the EMT term seems to be upregulated both in the normal (rank 3) and the cancer derived fibroblasts (rank 1). How do the authors explain this finding?
3. "In a word, the AT2 cell interacts specifically with the myeloid cell subset via the LGALS9 receptor, but that AT2-like cells interact with myeloid cells via ANXA1, FN1 and MDK, and imply that these ligand-receptor interactions may be a new immune checkpoint and potential immunotherapy target against LUAD." (line 385-388) - Very confusing sentence that needs editing. If I understood correctly,

the authors imply that they found that AT2-like cells interact with myeloid cells via the mentioned receptor/ligand pairs, which might be a new immune checkpoint and potential immunotherapy target. This conclusion cannot be drawn from the data at hand. Expression of receptor-ligand pairs on different cell types does not necessarily mean interaction between those cell types. The authors need to state more clearly that this is speculation that needs to be validated.

4. "Tumor ECs were overexpressed in all four histological stages tumors" (line 157). Wrong term, cells cannot be overexpressed.

We appreciate that the reviewers carefully read the manuscript, and thank the reviewers for their constructive criticism, which allowed us to improve our manuscript. We have added computational analyses and reorganized several figures (Reviewer #1). We also revised both the text and figures (updated numeration is used in this response) that address all reviewers' concerns, including a full discussion of the limitations related to our sample collection and analyses (Reviewer #2). We respond to each reviewer below:

Reviewer #1 (Remarks to the Author):

Reviewer 1 (2nd round review)

1. In figure 1B~D: presentation of Cluster info and then Normal through IA tumor in the same format and preferably side by side will clearly show the change of cell type abundance in the UMAP space. (Suggestion)

Author's Responses: We thank the reviewer for this suggestion. We made changes to the figures which now show the four stages in the UMAP space side by side in Figure 1D.

2. Figure 1G in the original submission (revised S5B) along with stats (revised S5C) clearly demonstrate the alterations in immune cell dynamics. Current Figure 1E can be supplementary. (Suggestion)

Author's Responses: We thank the reviewer for suggestion. We have moved Figure 1E to Figure S5A in the supplementary.

3. Individual patient data in the revised figure S5A is incomprehensible. Different format or reordering by analysis group (normal, AAH, AIS, MIA, IA tumor) will increase the visibility.

Author's Responses: We apologize that the data looked incomprehensible. There was too much of information to present in this figure. In the second revision we also present the data in a table format in the revised Table 3.

4. Revised line 116-117, where is the matching data?

Author's Responses: We have removed this sentence.

5. Revised Table 5 is redundant with Table 4. Use of the epithelial cell type marker genes (DEGs for Basal, Ciliated, Club, AT2 like, AT2, AT1) will be better to demonstrate AT2 likeness. Also figure format presentation of Table 5 seems to be more suitable along with revised Figure S6.

Author's Response: Thank you for your suggestion, we have moved table 5 in the revised manuscript along with revised Figure S6 in line 127.

6. Revised figure S6: demonstration of more markers (for Basal, Ciliated, Club, AT2 like, AT2, AT1) will be more convincing.

Author's Responses: We have revised Figure S6 and included additional markers as suggested for Basal, Ciliated, Club, AT2 like, AT2, AT1 cells according to Travaglini et al's recent publication (ref 17).

7. Results section 2 and revised figure2 focus on the normal vs. tumor analysis, not on the transitions at normal>>AAH>>AIS>>MIA>>stage IA LUAD. The normal-tumor analysis is mostly reiteration of previous publications.

Author's Responses: There are a number of recent publications focusing on the normal-tumor analysis of lung stromal cells and immune cells in microenvironment, so we performed our analysis in a similar fashion, to demonstrate that results are in agreement with previous findings. We think this is a validation of our dataset for those cell populations, and hence further strengthen our more unique findings related to the epithelial cells. We have moved this part to section 4 in lines 288-365.

8. Lines 216-218: Description of the Figure S13 needs to be more specific. Gene expression levels for CD4⁺ T, T follicular helper, Activated DC, and Granulocyte have a negative association where as those for plasmacytoid DC, CD141⁺ DC, and DC have a positive association with the survival period.

Author's Response: We have revised this sentence in the manuscript: "In fact, we found that the levels of seven cell subtypes, including CD4⁺ T, T follicular helper, activated DC, and granulocyte have a negative association, while other cell subtypes for plasmacytoid DC, CD141⁺ DC, and DC have a positive association with the survival period" in lines 360-363 in the revised manuscript.

9. Figure 3D is redundant with Supplementary Figure S14B.

Author's Response: Thank you for your suggestion, we have moved Figure 3D into Figure S10, and removed Figure S14B in the revised manuscript.

10. Figure 3E: Grouped (AAH>>AIS>>MIA>>stage IA LUAD) figure representation would be easier to comprehend.

Author's Response: Thank you for your suggestion, we have revised Figure 3E to 2D in the revised manuscript, and added Figure 2E to calculate CNV events present in tumor cells at different stages.

11. Figure 3G: Does IA LUAD include tumors with a lepidic growth pattern?

Author's Response: We performed immunofluorescence staining on different histologic stage of LUAD samples in Figure 3G (Revised Figure. 2G), to validated that canonical AT2 cell marker gene expression significantly decreased during cancer progression. And each staining in panels come from three samples. We repeated immunofluorescence staining for IA stage LUAD tissues (Patient 1 Papillary; Patient 2 Lepidic; Patient 3: Solid) in independent patients with similar results. We also added the predominant histological subtypes of IA stage LUAD patients in Table 1.

12. Figure 4A: The opposite direction of AAH and AIS vs. MIA and IA from the normal AT2 cells in the trajectory contradicts the authors' claim of gradual changes from normal to AAH>>AIS>>MIA>>stage IA LUAD. Why pseudotime 0 is assigned to the IA stage tumor cells?

Author's Response: Thank you for your suggestion and we apologize for an unintended mistake. We simply did not assign the appropriate time 0 for the pseudotime analysis, so that it was randomly assigned to one end of the plot, which happen to be the IA stage tumor cells. We have revised the figure and its legend as suggested in Figure 3A. Please note that this does not change the conclusion.

13. Line 392-395: Table 7 is the list of antibodies used and statistical tests were performed between pan-CK+ vs. pan-CK- cells, not between the tissue groups (Normal, AAH, AIS, MIA, stage IA LUAD) in the Supplementary Figure S23. How come these results confirmed that infiltration of immune cells increased with LUAD progression?

Author's Response: We thank the reviewer for highlighting this point. We have provided immunofluorescence statistics for Figure. 5D in Figure. S23. The results shown that infiltration of immune cells increased with LUAD progression. The statistical tests were performed between pan-CK⁺ vs. pan-CK⁻ cells, shown that PD-1 was high expression for pan-CK⁻ cells in AAH/AIA stage, while inversely PD-1 was high expression for pan-CK⁺ cells in MIA an IA stages in Figure. S24 (original Figure. S23). This is in line with a similar study showing that PD-1 could be expressed in tumor cells and could activate mTOR or Hippo signaling pathway, therefore facilitating tumor proliferation. We have revised the description in lines 390-399.

Reviewer #2 (Remarks to the Author):

I appreciate the efforts that the authors made to address the reviewers' questions and comments. I think the data generated are valuable for the field and most of the analyses are reasonable. The authors should be congratulated for doing such an important and challenging study. The comments from me and the other reviewers are mainly that many findings are overstated. Study on early carcinogenesis is difficult. There are a lot of limitations of the current study and many other previous studies, as exemplified below.

1. Profound heterogeneity of the lung cancers and precancers in different patients. This may be the main reason why many studies get different and even contra-indicatory results. Some recent studies such as PMID: 33083004; PMID: 33571124; PMID: 33976164 for example, should be discussed in the context of the findings from the current study.

Author's Response: Thank you for your suggestion. We have added the studies in the revised manuscript. In the revision, we have added the three references in the result as "Ref 18, Ref 48 and Ref 61" in lines 120, 326 and 416. These three studies attempted to decipher ecosystem of LUAD or immune evolution, and they have similar results with our study in result section. In the revision, we have added new discussion in line 414-416 as "Consistent with recent findings (Table 6), these alterations in stromal and immune populations cooperatively transformed immune-competent tissues into an immune-suppressive TME during LUAD progression"

2. The challenges of getting single cells from these early-stage diseases. As the authors mentioned, parried analyses could not be done due to small number of cells from single samples. For single

cell seq studies on cancers, people argue that samples with less than 3000 cells are not informative and should be removed. Some of the samples have way less cells than that. If the sample size is very big, this may not be a major issue since these could be balanced out. But given the small sample size, some findings could be driven by outliers.

Author's Response: Thank you for your suggestion and we agree with this comment. In the clinical setting, it is very difficult to obtain large number of cells (more than 3000) from patients in AAH and AIS stages due to the nature of the disease. Having more cells definitely improves the confidence, but to date there is no community consensus on the minimal cell numbers per sample. In fact, Tirosh et al have previously identified distinct tumor microenvironmental patterns using just low hundreds single metastatic melanoma tumors isolated per patient (ref 67). So single cell data obtained from small sample size can still be meaningful as long as the results are supported by statistical tests. Nonetheless, we have added relevant sentences regarding the limitation of our study in the discussion section in lines 466~468.

3. The controversial diagnosis of these entities: I don't think using two pathologists and focusing on GGOs will solve the problem of diagnosis of AAH, AIS and MIA. These have to be diagnosed by examining the whole section and many pathologist experts advocate for FFPE only and WHOLE section. There is no good solution at this time for such studies and the authors did the best they can. But you should discuss about this technical issue as the limitation for the field rather say you have solved the problem by focusing on GGO and using two pathologists.

Author's Responses: We thank the reviewer for highlighting this limitation. We have added relevant sentences in the discussion section as follow: "Third, it is worth noting that the tissue samples used for single cell sequencing and pathological diagnosis were different parts of the same surgical specimens, so there might be potential differences in their biological complexities of these parts" in lines 469-472.

The authors should clearly point out these limitations so future readers don't take these data interpretation as the "truth". These challenges will also be valuable for the whole field to brainstorming.

Reviewer #3 (Remarks to the Author):

The authors have responded to each of the reviewers' comments and have made changes in the manuscript accordingly. Most notably, the mouse data, plasma data, and FACS data were removed, while important patient data, including analysis of driver mutations, and immunofluorescence staining was added. The manuscript now has a clearer focus on the progression of human early-stage LUAD epithelial cells and the contributions of the tumor microenvironment. I agree with the authors' comment that the findings "are exciting but exploratory, and are therefore restricted to hypothesis generation." However, due to the paucity of published human early-stage LUAD data and the difficulties associated with acquiring early-stage patient samples. I think it will be important to publish this study and make it available to the research community after some minor revisions.

Comments:

1. With the normal cells present in the pseudotime analysis in figure 4A, the data does not convincingly show a linear progression from normal to AAH, AIS, MIA, and stage IA. Also, the authors did not update the main text after including the normal cells. In the heatmap in 4B, it looks like MIA cells, not stage IA cells, make up the right part of the heatmap. As another reviewer brought up, this linear relationship cannot be assumed and I wonder if this type of trajectory analysis is appropriate for this dataset, or if it should be left out. At the very least, the authors should comment on the limitations of using Monocle trajectory analysis on this dataset and explain why the normal cells line up the way they do in figure 4A.

Author's Responses: We thank the reviewer for highlighting this point. The normal cells were obtained from tumor-adjacent normal tissue, which are likely a heterogeneous population, and can skew the trajectory that leading to mis-interpretation of the results, so we decided not to include it. We agree with the reviewer's comment that the trajectory analysis has its limitations, so we added the comment in the discussion section in lines 475-480 as follow: "Beyond the outlined technical challenges, trajectory inference analysis also has its limitation. The field is still maturing, and the complexity of the underlying topology could be underestimated. Therefore, future studies on larger cohorts and utilizing tools designed to handle increasingly complex biological features are needed to dissect the evolutionary trajectory of early LUAD carcinogenesis and its underlying molecular mechanisms."

We apologize for a mistake in heatmap Figure 4B (revised Figure. 3B) as we accidentally swapped the labels in the legends. As you can see from Figure 4A (revised Figure. 3A), the stage

IA cells is at the end of the pseudotime plot, and it was also in the right part of the heatmap. As we were producing the figure, the plotting script sorted the labels by alphabetical order as default setting, so the color assigned to the cells in MIA stage was darker than those in IA stage. We made an unintended mistake by rearranging the text of the legends in the order of clinical stages (which MIA comes before IA) while putting the figures together. We thank the reviewer for pointing out the mistake and have revised the legend.

2. “GSEA comparing fibroblasts from normal and tumor tissues showed that cancer-derived fibroblasts were associated with the epithelial-mesenchymal transition (EMT) and with strong IFN- γ and IFN- α responses¹⁵ (Fig. 2F)” (line 179-181) – Actually, the EMT term seems to be upregulated both in the normal (rank 3) and the cancer derived fibroblasts (rank 1). How do the authors explain this finding?

Author’s Responses: We revised the sentences as follow: “GSEA comparing fibroblasts from normal and tumor tissues showed that cancer-derived fibroblasts were associated with the oxidative phosphorylation and with strong IFN- γ and IFN- α responses.” in lines 323-325.

3. “In a word, the AT2 cell interacts specifically with the myeloid cell subset via the LGALS9 receptor, but that AT2-like cells interact with myeloid cells via ANXA1, FN1 and MDK, and imply that these ligand-receptor interactions may be a new immune checkpoint and potential immunotherapy target against LUAD.” (line 385-388) – Very confusing sentence that needs editing. If I understood correctly, the authors imply that they found that AT2-like cells interact with myeloid cells via the mentioned receptor/ligand pairs, which might be a new immune checkpoint and potential immunotherapy target. This conclusion cannot be drawn from the data at hand. Expression of receptor-ligand pairs on different cell types does not necessarily mean interaction between those cell types. The authors need to state more clearly that this is speculation that needs to be validated.

Author’s Responses: We agree with this comment and have removed this sentence from the manuscript.

4. “Tumor ECs were overexpressed in all four histological stages tumors” (line 157). Wrong term, cells cannot be overexpressed.

Author's Responses: We apologize for the inappropriate wording. We have changed the sentences as follow: "Tumor ECs were observed in the tumor tissues of all four histologic stages" in lines 300-301.

REVIEWERS' COMMENTS

Reviewer #1 (Remarks to the Author):

Please check irregularities and minor errors.

Regarding Figure S4: It is not clear what the colors indicate. Green color means wild type call? What is the purple color? Please revise the figure or legend for the clarification.

Regarding lines 270-273 (Figure S13B and S14): I don't find a gradual decrease in Ki67 expression. Perhaps the authors meant "no significant changes during progression" ?

Regarding Figure 4 D and E: Color codes of Normal fibroblast and Fibroblast-like cells are switched.

Regarding Figure 4F: "Epithelial-Mesenchymal Transition" is positively enriched in both Normal and Tumor? Is it possible?

Line 360: TME type-O

Figure 5D: Is PD-L1 (Green signal) ubiquitously expressed in the Normal lung tissues? Is it a background?

Reviewer #2 (Remarks to the Author):

The authors have adequately addressed my comments.

Reviewer #3 (Remarks to the Author):

The authors have addressed all reviewer concerns and the paper and its data should be published in Nature Communications.

REVIEWERS' COMMENTS

Reviewer #1 (Remarks to the Author):

Please check irregularities and minor errors.

Regarding Figure S4: It is not clear what the colors indicate. Green color means wild type call? What is the purple color? Please revise the figure or legend for the clarification.

Author's Responses: We thank the reviewer for this suggestion. Green color cells mean wild type of the given patients. Purple color cells mean that driver gene mutations were detected in the given patients. We have revised the legend in figure S3 as "The labels indicate tumor identifiers. Yellow cells represent mutated genes. Green color cells mean wild type of the given patients. Purple color cells mean that driver gene mutations were detected in the given patients. Grey color cells mean no specimens available."

Regarding lines 270-273 (Figure S13B and S14): I don't find a gradual decrease in Ki67 expression. Perhaps the authors meant "no significant changes during progression" ?

Author's Responses: We have revised this sentence in the manuscript: "Interestingly, the staining intensity in our study showed an increase in Ki67 expression (gray) from normal to AAH/AIS stage, however, there was no significant changes during progression" in lines 247-249 in the revised manuscript.

Regarding Figure 4 D and E: Color codes of Normal fibroblast and Fibroblast-like cells are switched.

Author's Responses: Thank you for your suggestion and we apologize for an unintended mistake. We have switched the color codes of Normal fibroblast and Fibroblast-like in Figure 4e.

Regarding Figure 4F: "Epithelial-Mesenchymal Transition" is positively enriched in both Normal and Tumor? Is it possible?

Author's Responses: We thank the reviewer for highlighting this point. The normal cells were obtained from tumor-adjacent normal tissue. As the reviewer has mentioned in the first revision, some normal cells indeed contribute to AT2-like clusters. This could possibly be two reasons: Travaglini et al's have recently uncovered two clusters of AT2 cells of normal lung cell atlas. One cluster expressed higher levels of canonical AT2 markers (SFTPA1, SFTPC and ETV5). The other one, AT2-signalling cells, could be a rare population of alveolar stem cells, which is similar to Wnt-active subpopulation of AT2 cells (Nature 2020 Ref.17). This result provided a consistent phenotype with our outcome. On the other hand, these tumor adjacent normal tissues might contain small fraction of tumor cells due to biological/technical challenges during sample collection. So Epithelial-Mesenchymal Transition could be enriched in both normal and tumor tissues.

Line 360: TME type-O

Author's Responses: We apologize for the inappropriate wording. We have changed the "TEM" to "TME" in lines 330.

Figure 5D: Is PD-L1 (Green signal) ubiquitously expressed in the Normal lung tissues? Is it a background?

Author's Responses: We thank the reviewer for highlighting this point. Seven colors were merged in each figure in Figure 5d, so there would be some deviation. We can see a separate stained figure as follow (Figure 1), PD-L1 had low expression level in normal tissues. we also find similar statistical results in figure S22.

Figure 1. Staining of PD-L1⁺ cells (green) on normal tissues.

Reviewer #2 (Remarks to the Author):

The authors have adequately addressed my comments.

Reviewer #3 (Remarks to the Author):

The authors have addressed all reviewer concerns and the paper and its data should be published in Nature Communications.